behaviour, ecology

sea turtle 'lost years', *Chelonia mydas*, satellite telemetry, Sargasso Sea, sea turtle nursery, sea turtle developmental habitat

**Author for correspondence:**
Katherine L. Mansfield
e-mail: kate.mansfield@ucf.edu

# First Atlantic satellite tracks of 'lost years' green turtles support the importance of the Sargasso Sea as a sea turtle nursery

Katherine L. Mansfield[1], Jeanette Wyneken[2] and Jiangang Luo[3]

[1]Department of Biology, University of Central Florida, Orlando, FL 32816, USA
[2]Department of Biological Sciences, Florida Atlantic University, Boca Raton, FL 33431, USA
[3]Department of Marine Ecology and Society, Rosenstiel School of Marine and Atmospheric Science, University of Miami, Miami, FL 33149, USA

KLM, 0000-0002-6568-2861; JW, 0000-0002-9441-249X; JL, 0000-0001-8068-0496

In-water behaviour and long-term movements of oceanic-stage juvenile sea turtles are not well described or quantified. This is owing to technological or logistical limitations of tracking small, fast-growing animals across long distances and time periods within marine habitats. Here, we present, to our knowledge, the first long-term offshore tracks of oceanic green turtles (*Chelonia mydas*) in western North Atlantic waters. Using a tag attachment technique developed specifically for young (less than 1 year old) green turtles, we satellite-tracked 21 oceanic-stage green turtles (less than 19 cm straight carapace length) up to 152 days using small, solar-powered transmitters. We verify that oceanic-stage green turtles: (i) travel to and remain within oceanic waters; (ii) often depart the Gulf Stream and North Atlantic Subtropical Gyre currents, orienting towards waters associated with the Sargasso Sea; (iii) remain at the sea surface, using thermally beneficial habitats that promote growth and survival of young turtles; and (iv) green turtles orient differently compared to same stage loggerhead turtles (*Caretta caretta*). Combined with satellite tracks of oceanic-stage loggerhead turtles, our work identifies the Sargasso Sea as an important nursery habitat for North Atlantic sea turtles, supporting a growing body of research that suggests oceanic-stage sea turtles are behaviourally more complex than previously assumed.

## 1. Background

Sea turtle life history is defined by ontogenetic shifts in habitat use as turtles mature (reviewed by [1–4]). All sea turtle species start life moving from terrestrial nests through to coastal waters as hatchlings. Most species then quickly move to oceanic marine environments as hatchlings and post-hatchlings [1,5]. With the exception of flatback turtles (*Natator depressus*; [6]), most hard-shelled (cheloniid) species are assumed to migrate from natal neritic waters to oceanic nursery areas (waters greater than 200 m in depth), eventually returning to coastal waters as larger juveniles where they grow to maturity and through adulthood [1,5]. However, few in-water empirical data are available on the early movements and behaviour of these cheloniid species, aside from loggerheads (*Caretta caretta*) and some green turtles originating from rookeries in the North or South Atlantic (e.g. [7–9]). Juvenile North Atlantic green turtles return to coastal waters once they reach approximately 20 cm straight carapace length (SCL) [10]; yet, the turtles' preceding oceanic period remains unknown as empirical observations of the turtles' long-term movements, behaviour and habitat use are lacking [11–13].

Information derived from opportunistic offshore sightings, short-term (hours in duration) visual tracking from natal beaches and laboratory work with hatchling loggerhead turtles (*Ca. caretta*) resulted in long-held hypotheses regarding the early sea turtle behaviour (e.g. [3,11,14–17]). These hypotheses and assumptions include (adapted from Mansfield *et al.* [7]):

(i) **hypothesis 1:** upon hatching, sea turtles swim offshore to oceanic waters off the Continental Shelf (greater than 200 m in depth) where they remain for several years [3,11];

(ii) **hypothesis 2:** once offshore, these oceanic-stage turtles travel, or drift, passively within ocean currents (specifically those currents associated with the North Atlantic Subtropical Gyre (NASG), in the North Atlantic Ocean) for the duration of their offshore life stage [11,15]; and

(iii) **hypothesis 3:** oceanic-stage turtles remain at the sea surface while offshore where they likely associate with floating habitats such as *Sargassum* and other flotsam [11,14–16,18].

These hypotheses were historically assumed to apply across species, including oceanic-stage green turtles. Prior in-water studies on the early lives and behaviour of oceanic green turtles focused on initial offshore orientation [19], documentation of predation in near shore waters [20] and diving ontogeny [21]. These studies indicate that upon entering the ocean, hatchling and post-hatching greens remain near the sea surface and after their first day in the water, turtles alternate between diurnal periods of active swimming, followed by periods of night-time rest during their first week at sea [1,22,23]. Similar to oceanic-stage loggerhead turtles, green turtles associate with flotsam and floating *Sargassum* [11,18,24,25]. Trace element and stable isotope data from *Chelonia mydas* epidermal samples [12,13] are consistent with the hypothesis that young green turtles inhabit oceanic habitats during their first years of life and suggest that some Atlantic green turtles forage in east-northeastern Atlantic waters and the Sargasso Sea, the region contained within the currents making up the NASG [26]. Predictive models examining early loggerhead and green dispersal from western Atlantic rookeries hint at a combined use of both the currents associated with NASG and the Sargasso Sea following their initial dispersal from their natal beaches along the Atlantic United States (US) coastline [5,27]. However, the nursery habitats used by young green turtles during their first years at sea are not yet fully described. This knowledge gap is largely owing to technological limitations and logistic obstacles that constrained telemetry studies to larger turtles, or limited the spatial scope of studies to near shore or opportunistic observations [7,28]. Understanding the early behaviour and identifying early developmental habitats is critical for the successful management and conservation of this historically imperiled species [29–31].

Mansfield *et al.* [28] developed methods to attach solar-powered platform transmitting terminal (PTT) bird tags to oceanic-stage loggerheads in the western Atlantic, contributing the first satellite tracks of any 'lost year' turtle. The loggerhead turtles in this study [7,28] were tracked up to 220 days, resulting in the first characterization of very young turtles' movements, dispersal routes and habitat use for early oceanic life stages [7]). This work validated the long-standing hypothesis that the turtles were indeed oceanic (hypothesis 1; remaining off of the Continental Shelf or in waters greater than 200 m depth [3,11]). These turtles remained at the sea surface and associated with the NASG and its mesoscale eddies, validating hypothesis 3 [7,14–18]. However, Mansfield *et al.* [7] also observed that some turtles departed the currents associated with the NASG, displaying directional movements into the interior of the NASG and the Sargasso Sea—an unexpected deviation from

the long-held hypothesis (hypothesis 2; [11,15]) that Atlantic loggerheads travel around the Atlantic basin remaining within NASG currents until recruiting to near shore developmental habitats as large juveniles [11]. Based on this prior neonate loggerhead tracking work, we now question whether oceanic green turtles will behave similarly to loggerheads and remain within oceanic waters, at the sea surface, solely within the currents of the NASG or whether they too, will enter into the Sargasso Sea.

Wild-caught oceanic-stage green turtles in the Gulf of Mexico are known to associate with *Sargassum* spp. and other flotsam and are observed to occur at the sea surface [9,18,24]; however, turtles in the Gulf of Mexico probably originate from several rookeries including Florida and many outside of the US (Mexico, Costa Rica, the Caribbean) [32]. These turtles encounter very different current regimes and oceanographic features than turtles hatchling along the Atlantic US coast, and thus their dispersal patterns and habitat selection are not easily compared to loggerheads originating from Atlantic US rookeries. The telemetry work detailed by Putman & Mansfield [9] occurred several years after the track data presented in this study and focused on the long-held assumption that oceanic-stage turtles are 100% passive drifters for the entirety of their oceanic life stage. Tracked oceanic-stage green turtles captured in the north or northwest the Gulf of Mexico actively swam (were not passive drifters) and exhibited species-specific directed orientation to possibly leave the Gulf of Mexico to the southeast, entering the Straits of Florida and the current that becomes the Gulf Stream in the western North Atlantic [9]. Based on these insights, dispersal models [27] and prior work with loggerheads in the North and South Atlantic oceans [7,8], we predict that Atlantic green turtles are behaviourally more complex during their first years than previously hypothesized.

Building on previous studies (Mansfield *et al.* [7,8,28]), our objectives were to: field-test a new, species-appropriate satellite tag attachment method for laboratory-reared oceanic-stage green turtles; characterize their in-water movements; and identify habitats used to better understand the offshore features that probably define early green turtle nursery habitats, as well as potential areas of risk to these juvenile turtles during their 'lost years' oceanic developmental period. Following Mansfield *et al.* [7], we test whether young (less than 1 year old) juvenile green turtles (i) remain off the US Continental Shelf within oceanic waters; (ii) remain exclusively within the currents associated with the NASG; and (iii) remain at the sea surface for the duration of their tracked movements. Finally, we compare our results to tracks of oceanic-stage loggerheads [7,33] that hatched from the same beaches, were raised to the same age under similar conditions, released within the same region as the green turtles in this study to (iv) test for differences in orientation and habitat use among species and better identify critical early sea turtle developmental habitat in the Northwest Atlantic Ocean.

## 2. Material and methods

### (a) Study animals

Hatchling green turtles were collected upon emergence from natural nests in Boca Raton, Florida USA (26.42° N, 80.03° W). Turtles were raised at the Florida Atlantic University (FAU) Marine Laboratory following protocols modified from Stokes *et al.* [34] and described by Mansfield *et al.* [28] for loggerhead

turtles. Turtles were housed in flow-through seawater tanks maintained at an average 25.7°C (±2.1°C), fed daily an in-house manufactured food at 8–11% of their bodyweight and provided with a 12 L : 12 D photocycle. Turtles were laboratory reared to greater than 300 g in weight and approximately 12.0 to 18.6 cm SCL (3–9 months of age [28]).

## (b) Transmitter attachment

The tag attachment method developed for loggerhead turtles [7] failed in green turtles—tags were shed from the turtles' shells within 7–14 days of attachment; the loggerhead method did not stick to the green's shells, regardless of the degree of shell preparation. To ensure a successful tag attachment method for this study, neonate green turtle carapaces were cleaned with a terry cloth towel and dilute 2% chlorhexidine gluconate solution (per manufacturer instructions for disinfecting skin) until no dark material could be wiped/buffed from the turtles' scutes. The first three vertebral scutes and medial aspects of the first two costal scutes were very lightly scored with 100–200 grit sandpaper (figure 1a) to remove loose flaking keratin, then wiped with the dilute chlorhexidine solution and allowed to dry. Three marine urethane adhesives (3M 5200™, 3M 5200 Fast Cure (FC)™, and 3M 4200™) met our criteria for successful tag attachment (per [7,28]): tag attachment durations of greater than two months; efficacy for use in the field (e.g. minimal adhesive set time) and no adverse effects of the attachments on turtle growth and behaviour. Prior to this study, we observed turtles in controlled settings with and without attachments; this green turtle attachment method exceeded tag retention goals (greater than three months in the laboratory) and did not affect diving, surfacing, swimming, foraging, body attitude in the water or turtles' ability to rest at the surface. Turtles greater than 300 g met our tag attachment criteria in controlled laboratory conditions (smaller turtles did not all meet the criteria); therefore, turtles greater than 300 g (and greater than 12 cm SCL) were our target size for this study.

We affixed Microwave Telemetry's PTT-100 9.5 g solar-panelled satellite bird tags (pressure proofed for a marine environment) to 21 laboratory-reared green turtles using 3M 5200™ (figure 1b) or 3M 5200 FC™ adhesives. Tags were temporarily held in place with a small strip of Mylar™ tape extending from the costal scutes and across the tag while the adhesives cured. The tag attachment was allowed to dry in the air for 2–6 h and a fairing was built around the tag with additional adhesive to help ensure a hydrodynamic shape to the attachment per Jones et al. [35]. As the adhesive cured, we added coloration (e.g. indelible marker figure 1c) to better match the colour of the turtles' shells. We observed the tagged turtles in seawater tanks to ensure all turtles acclimated fully to the attached tags prior to field release.

This novel satellite tag attachment method was used successfully to track all 21 oceanic-stage green sea turtles in the Atlantic Ocean in 2012 and 2013. All turtles were released within the Gulf Stream in *Sargassum* habitat (approximately 26.7° N latitude, and between 79.4° and 79.9° W longitude) offshore of their natal beaches in southeast Florida.

## (c) Turtle movements, habitat use and orientation

Transmitter location and sensor data including tag charge (V) at time of transmission were obtained via the Argos system. Data were filtered for location accuracy following Mansfield et al. [7] using Location Codes 3–0, A and B, and filtered locations were interpolated to 6 h intervals using modified piecewise Bézier methods (after: http://ljensen.com/bezier/; [36,37]). Track data were overlaid with bathymetric gridded global relief data (ETOPO2 v. 2) and composite sea surface temperature (SST) data derived from the global hybrid coordinate ocean model (HYCOM + NCODA global 1/12° analysis; 7 km resolution)

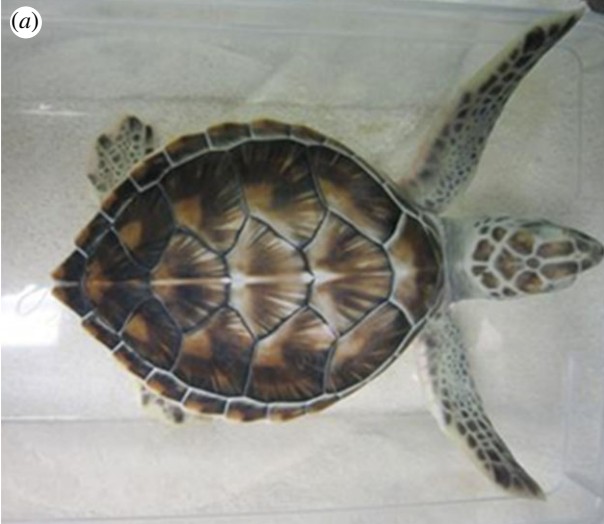

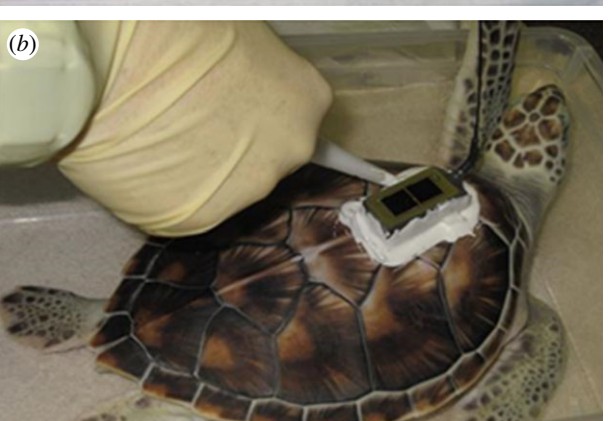

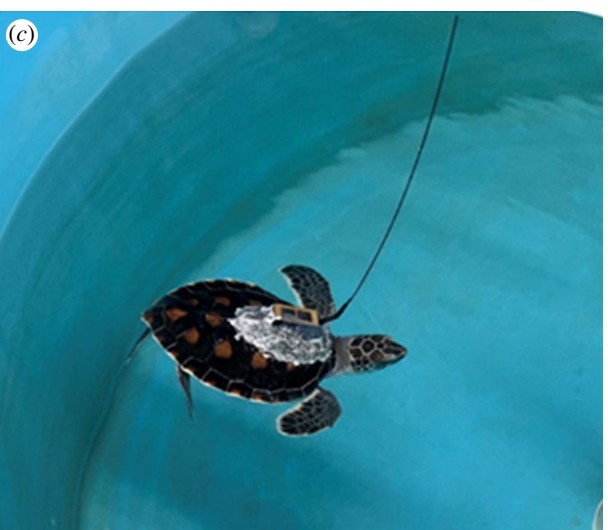

**Figure 1.** (a–c) Cleaned anterior carapace and 3M 5200 tag attachment method used to satellite tag young green turtles (less than 19 cm SCL). (a) A juvenile green turtle's shell was first cleaned then the first three vertebral scutes and medial aspects of the first two costal scutes were very lightly score with 100–200 grit sand paper. The cleaned area, following the removal of any dust from the scutes, appears lighter than the rest of the shell. (b) A base layer of 3M 5200 Fast Cure™ adhesive was applied to the bottom of the tag prior to placement on the shell. (c) The fairing and colouring before permanent marker are shown on a turtle in its tank. Photo credits: K. Mansfield (a,b) and J. Wyneken (c). (Online version in colour.)

and with MODIS (9 km resolution) using the satellite tracking and analysis tool (STAT; [7,38]).

We conducted kernel home range analyses on all green turtle tracks combined using a probabilistic fixed kernel density model using the daily positions. We used Worton [39] algorithms to

identify key areas of habitat use (%, ranging from 95% likelihood of occurrence to 50–25% core use areas). We also conducted the same kernel analyses on the combined filtered track data from 17 oceanic-stage loggerhead turtles (derived from data presented in [7,33]) to identify areas of high habitat use by oceanic-stage sea turtles in the western North Atlantic. Finally, we conducted kernel analyses on green and loggerhead tracks (separately and combined) falling within the boundaries of the Sargasso Sea (figure 2a; as defined by Laffoley et al. [26]).

We estimated the turtle orientation vector ($V_t$) for each 6 h interval position by the differences between the vectors of turtle position ($V_p$) and the ocean current vector ($V_h$) from HYCOM global model: $V_t = (V_p - V_h)$. Using the 6 h heading data, we estimated the mean orientation for each tagged turtle for each day during the tracking period. The frequency distributions of daily turtle orientation were plotted using rose diagrams [40] and were tested statistically for uniform distribution using both parametric Rayleigh's z-test and nonparametric Watson's $U^2$-test as described in Zar [41].

To test whether oceanic-stage green turtles remained within the prevailing currents of the NASG or, like loggerheads, departed ocean currents and travelled to the Sargasso Sea (hypothesis 2), we compared track orientation vectors using Watson's $U^2$ nonparametric two-sample test [41]. A subset of all individual green turtles' tracks occurring south of latitude 37° N and west of longitude 72° W were pooled to test the differences in mean orientation against the same subset of satellite tracks from loggerhead turtles reported in Mansfield et al. [7]. This subset represents the spatial range within which both green and loggerhead tracks occurred (maximizing available track data), the northern limit of the NASG, and includes the region tested for loggerheads from Mansfield et al. [7] encompassing the artificial magnetic field locations used by Lohmann & Lohmann [17]—a region where laboratory-reared naive loggerheads oriented to the east northeast to theoretically remain within the NASG.

# 3. Results

## (a) Turtle movements

We remotely tracked 21 neonate green turtles for 10–152 days (mean: 65.9 ± 30.6 days s.d.; table 1) in the western Atlantic Ocean. Turtles largely remained in oceanic waters, off the Continental Shelf (greater than 200 m depth; figure 2b). Out of 1379 track days, turtles spent 93 track days within Continental Shelf waters (6.7% of total track time among all turtles). The majority of this time spent on the Continental Shelf was associated with the turtles' initial release; only one turtle (85514b) remained in Continental Shelf waters for the entirety of its track (figure 2b). Most turtles remained east of the Gulf Stream's western frontal boundary and well off the Continental Shelf (figure 2b; the exceptions were turtles 85512b, 85514b, and the last location transmitted from 117332b). However, the eastern edge or frontal boundary of the Gulf Stream did not constrain turtles and many travelled into the western Sargasso Sea from the Gulf Stream (figure 2b). Fifteen of the 21 green turtles tracked departed the Gulf Stream and Gyre currents. Fourteen of these turtles entered the western or northern Sargasso Sea region in the western Atlantic, while one turtle (117332b) travelled onto the Continental Shelf to the west of the Gulf Stream shortly before its tag ceased transmitting. Nine turtles departed the Gulf Stream south of or near Cape Hatteras, North Carolina USA and six turtles departed the Gulf Stream towards the Sargasso Sea from locations north and east of Cape Hatteras. The remaining turtles had not entered the Sargasso Sea prior to tag cessation. Loggerhead

track data from Mansfield et al. [7,26] are presented in figure 2c for comparison to the green turtle tracks in this study. Seven of 17 of these loggerhead turtles travelled out of the Gulf Stream and into the Sargasso Sea [7].

## (b) Habitat use and orientation

Argos location classes (LCs) combined with the tag charge data derived from the solar PTT sensor output suggest that the tracked turtles remained mostly at the sea surface (e.g. [7]). A total of 5505 messages were received from the turtles' tags; 64.9% were Argos LCs 3–1 (n = 25 were class Z and eliminated from the analyses owing to lack of associated location data; the remainder were LCs A and B). All tags maintained optimal mean charges throughout the duration of the track periods (4.04 V mean ± 0.10 V s.d., range: 3.2–4.4 V). This consistent optimal charging indicates that the tags were exposed to air (promoting communication with overhead satellites) and to direct sunlight (promoting efficient charging of the solar-powered tags; [7,28]). Ambient temperatures recorded by the tags' internal thermal sensors ranged from 12.1°C to 37.4°C (mean: 25.1° ± 2.7°C s.d.).

Kernel analyses of 21 green turtle tracks and 17 loggerhead tracks [7,26] were spatially biased to initial release location and Gulf Stream-driven dispersal (figure 3a–c). High use areas for each species were identified within the Gulf Stream waters immediately post-release (figure 3a,b), and within the waters of the western Sargasso Sea region to the east of the Gulf Stream (figure 3d,e). Combined loggerhead data from Mansfield et al. [7,26] and green turtle tracks show areas of high spatial use to the east of the Gulf Stream frontal boundary, within the western Sargasso Sea (figure 3c,f).

Orientation analysis indicated that 10 out of 21 tagged green turtles displayed statistically significant orientation, while 11 individuals had headings that were not statistically different from a uniform distribution (electronic supplementary material, table S1). The mean heading of the pooled subset of all individual green turtle tracks (n = 17) south of 37° N and west of 72° W is south (approximately 182°; figure 4a) and is statistically different from a uniform distribution ($U^2 = 0.4738$, n = 718, p < 0.001). Compared to oceanic-stage loggerhead turtles from Mansfield et al. ([7]; figure 2c), 12 out of 17 loggerheads displayed statistically significant orientation, and five were not statistically different from a uniform distribution (electronic supplementary material, table S2). Mean loggerhead heading for pooled tracks south of 37° N and west of 72° W was towards the north (approximately 013°, figure 4b) and was statistically different from uniform distribution ($U^2 = 0.3065$, n = 595, p = 0.002–0.005). Nonparametric (Watson's $U^2 = 0.509$, n1 = 718, n2 = 595, p < 0.001) two-sample tests indicate significant differences in mean orientations of oceanic-stage green and loggerhead turtles in the western North Atlantic (figure 4c,f; electronic supplementary material, tables S1 and S2). When we used only data from the 10 green and 12 loggerhead turtles with significant headings for comparison, the results are nearly the same (figure 4c,d; electronic supplementary material, tables S1 and S2) with greater statistical significance for the nonparametric two-sample test ($U^2 = 0.643$, n1 = 371, n2 = 321, p < 0.001).

# 4. Discussion

This study provides, to our knowledge, the first long-term satellite tracks for oceanic-stage green turtles using a novel satellite tag attachment method developed specifically for this species.

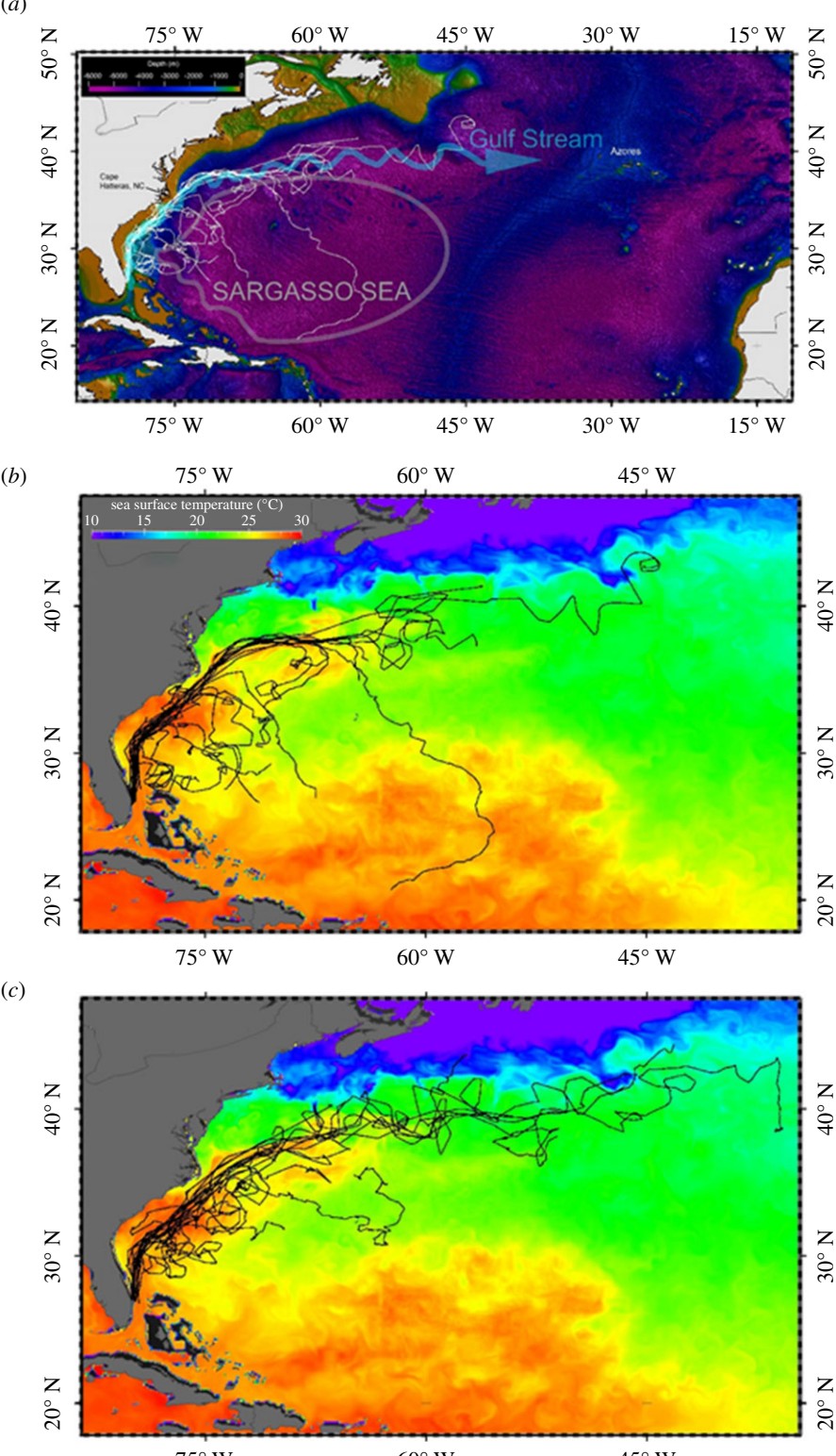

**Figure 2.** (*a–c*) Satellite tracks from 21 laboratory-reared neonate green sea turtles (13.9–18.6 cm straight carapace length) released in the western Atlantic Ocean are presented (*a*) in reference to the Gulf Stream and general boundaries of the Sargasso Sea [26]. The tracks (white lines) are overlaid on bathymetric gridded global relief data, ET0P02v2 showing the various routes taken by turtles leaving the Continental Shelf (light brown) and entering oceanic waters (blue and purple). (*b*) Composite SST data (with green turtle tracks overlaid in black) derived from satellite imagery databanks and the global hybrid coordinate ocean model (HYCOM + NCODA global 1/12° analysis; 7 km resolution). (*c*) Composite SST data with loggerhead turtle tracks overlaid in black [7,33] derived from satellite imagery databanks and the global hybrid coordinate ocean model (HYCOM + NCODA global 1/12° analysis; 7 km resolution). Loggerhead map (*c*) adapted from Mansfield *et al.* [7,33].

Our work also provides the first comparison of oceanic-stage movements and behaviour between two species originating from the same southeast US sea turtle rookeries. The green turtles tracked in this study, and the oceanic-stage loggerheads from Mansfield *et al.* [7] share several biological and ecological similarities: both are members of the Cheloniidae family, they share the same nesting beaches, and both species use the Gulf Stream in their initial dispersal from natal Atlantic beaches.

**Table 1.** Metadata for 21 laboratory-reared green turtles satellite tracked in this study, including: satellite tag identity (ID), SCL measured from the nuchal notch to the posterior-most tip of the supracaudal scutes, mass, body depth (the straight-line measurement of the highest point of the carapace to the deepest point on the plastron measured with Vernier calipers positioned parallel to the turtle body axis; after Wyneken [42]), sex (as determined by laparoscopic exam for a separate study), age, hatch and release dates, release locations and track durations.

| turtle ID | SCL (cm) | weight (g) | body depth (cm) | sex | age (d) | hatch date | release date | release latitude | release longitude | track duration (d) |
|---|---|---|---|---|---|---|---|---|---|---|
| 92585c | 18.1 | 790 | 7 | F | 247 | 7/30/2011 | 4/2/2012 | 26.783 | −79.900 | 100 |
| 117332a | 17.4 | 786.6 | 7.4 | F | 271 | 7/30/2011 | 4/26/2012 | 26.762 | −79.856 | 93 |
| 92588c | 17.2 | 793 | 7.7 | F | 268 | 8/2/2011 | 4/26/2012 | 26.762 | −79.856 | 102 |
| 85513b | 16.4 | 640.5 | 6.8 | F | 281 | 8/2/2011 | 5/9/2012 | 26.714 | −79.918 | 70 |
| 92584c | 16.5 | 705.9 | 7 | F | 268 | 8/15/2011 | 5/9/2012 | 26.714 | −79.918 | 10 |
| 85511b | 17.1 | 704.8 | 7.4 | F | 248 | 9/8/2011 | 5/13/2012 | 26.763 | −79.844 | 53 |
| 85512b | 18.6 | 799.8 | 7.39 | F | 248 | 9/8/2011 | 5/13/2012 | 26.763 | −79.844 | 83 |
| 85514b | 18.1 | 857 | 7.36 | M | 247 | 9/9/2011 | 5/13/2012 | 26.763 | −79.844 | 60 |
| 92586c | 15.1 | 517.9 | 6.17 | M | 200 | 10/26/2011 | 5/13/2012 | 26.763 | −79.844 | 58 |
| 92587c | 16.3 | 640 | 6.48 | M | 200 | 10/26/2011 | 5/13/2012 | 26.763 | −79.844 | 43 |
| 92590c | 16.6 | 559.7 | 6.64 | M | 200 | 10/26/2011 | 5/13/2012 | 26.763 | −79.844 | 83 |
| 117332b | 13.82 | 374 | 5.67 | F | 133 | 8/5/2012 | 12/16/2012 | 26.751 | −79.734 | 51 |
| 92584 | 13.76 | 404 | 5.68 | F | 133 | 8/5/2012 | 12/16/2012 | 26.749 | −79.701 | 36 |
| 92586 | 14.68 | 467 | 6.12 | M | 141 | 7/28/2012 | 12/16/2012 | 26.749 | −79.701 | 66 |
| 92590 | 15.16 | 487 | 5.75 | M | 141 | 7/28/2012 | 12/16/2012 | 26.751 | −79.734 | 56 |
| 85513 | 14.4 | 425.4 | 5.9 | F | 158 | 7/28/2012 | 1/2/2013 | 26.749 | −79.701 | 152 |
| 85514 | 13.9 | 442.4 | 6.4 | F | 158 | 7/28/2012 | 1/2/2013 | 26.749 | −79.701 | 46 |
| 85511c | 12.78 | 303.1 | 5.2 | F | 121 | 9/14/2012 | 1/13/2013 | 26.749 | −79.701 | 85 |
| 85512 | 13.04 | 333 | 5.45 | F | 121 | 9/14/2012 | 1/13/2013 | 26.749 | −79.701 | 63 |
| 92588 | 13.3 | 362.5 | 5.53 | F | 165 | 9/2/2012 | 2/14/2013 | 26.742 | −79.493 | 38 |
| 92587 | 11.99 | 315.9 | 5.39 | F | 166 | 9/2/2012 | 2/15/2013 | 26.742 | −79.793 | 36 |

## (a) Study animals and transmitter attachment

Oceanic-stage green and loggerheads turtles differ in carapace characteristics: green turtle shells are flatter and lack elevated vertebral keels characteristic of young, oceanic-stage loggerheads. The keratinous scutes on green turtle shells are thin, black to dark brown in colour and have a 'waxy' surface necessitating that we use a different tag attachment method than for the loggerheads [28]. We found that the urethane marine adhesives 3M 5200™ or 3M 4200™ (regular and FC) provided similar attachment durations in the laboratory and field compared to loggerheads [7]. While curing was slower than we would like (e.g. 2–6 h for regular 3M 5200™ or 3M 4200™, and 1–2 h for FC), these adhesives complete the curing process in seawater, thus potentially shortening the holding time prior to release—an advantage for field-based applications and tag deployments with wild-caught turtles. We do not recommend using the loggerhead tag attachment method described in Mansfield [28] for green turtles.

## (b) Turtle movements, habitat use and the importance of the Sargasso Sea

Our track data support the prediction that, like loggerheads, oceanic green turtles are more behaviourally complex than previously hypothesized. Our results support the long-held hypotheses that North Atlantic oceanic-stage green turtles, like loggerheads, inhabit oceanic habitats (hypothesis 1) and probably remain mostly at the sea surface during their early years at sea (hypothesis 3). However, green turtles in our study spent considerable time in the Sargasso Sea (figure 3d–f), a region inside of the NASG and the gyre currents. This leads us to reject hypothesis 2 as it was historically applied to all oceanic-stage loggerhead and green turtles.

Lopez-Castro [12] inferred that the Sargasso Sea is an important feeding habitat for juvenile green turtles based upon the signature of metals and isotopes in their tissues. Putman et al. [27] dispersal models predicted that some oceanic-stage green turtles and loggerheads will travel to the Sargasso Sea during their oceanic phase; however, this work assumed passive drift and did not factor in active swimming [9] or species-specific differences in orientation. Our study provides empirical evidence identifying Sargasso Sea waters as part of the Atlantic green turtle nursery habitat. Two-thirds of our tracked green turtles were within Sargasso Sea waters when their tags ceased to transmit, suggesting that many Atlantic green turtles use the Sargasso Sea as a nursery habitat. Here, we provide direct confirmation that oceanic-stage green turtles that initially disperse within the Gulf Stream are likely to leave this current and enter into the Sargasso Sea. Similar to Mansfield et al. [7], our data do not support the long-held hypothesis and assumption that oceanic-stage turtles will remain in the currents making up the NASG in a unidirectional developmental migration around the North Atlantic Ocean.

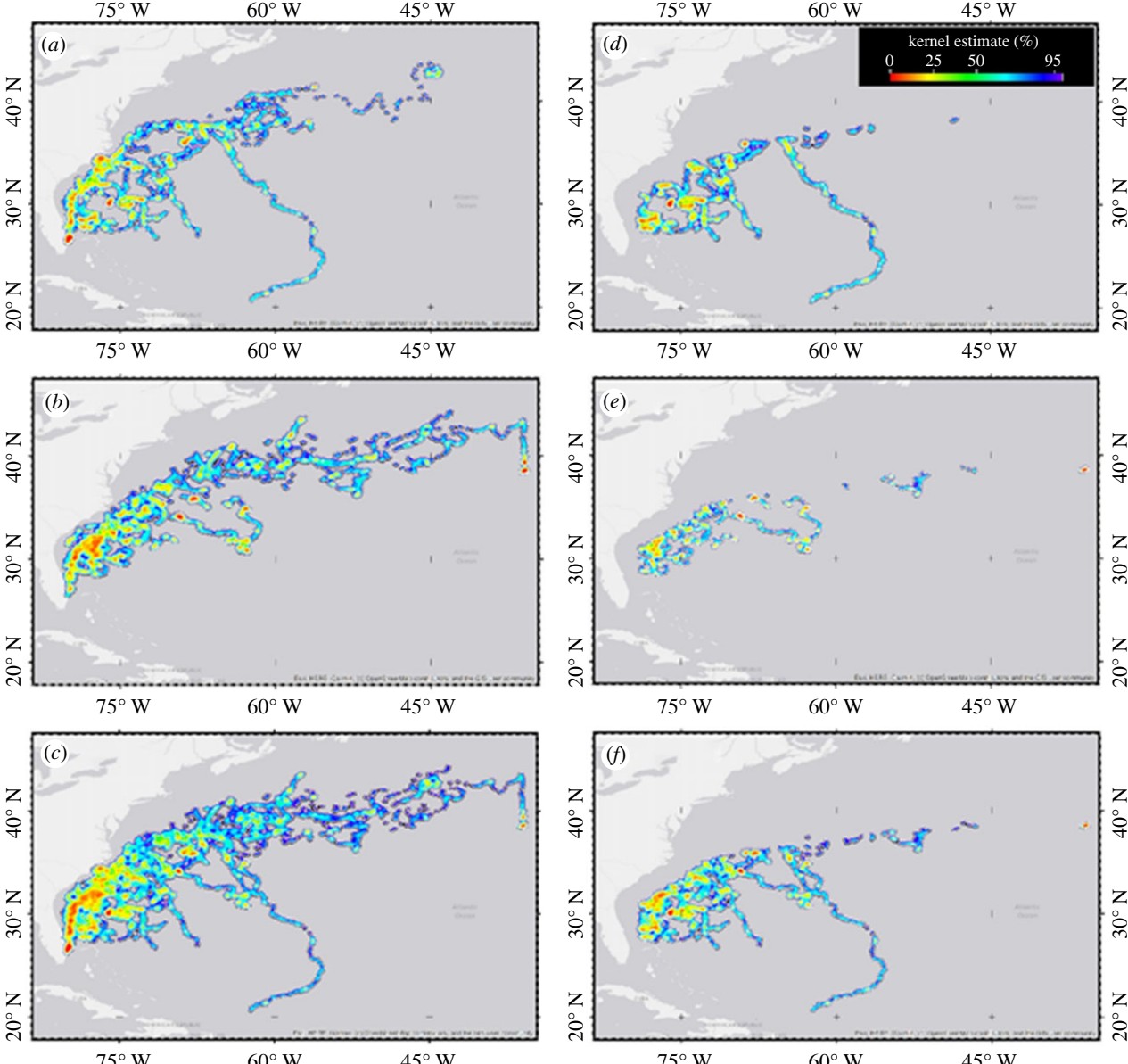

**Figure 3.** (a–f) Kernel analyses of (a) satellite tracks from 21 green turtles, and loggerhead tracks (b) from Mansfield *et al.* [7,33], and (c) all combined tracks (green and loggerhead) including initial tracks from Gulf Stream release off the southeast Florida coast. All turtles hatched from the same beaches and released in the same region (data from [7,33]). Kernel analyses of the subset of the green (d), loggerhead (e) and combined (f) species tracks occurring within the Sargasso Sea boundaries (per Laffoley *et al.* [26]). The 'warmer' the colour associated with the tracks used for the kernel estimates, indicates the higher density habitat use by turtles [39]. These data illustrate the importance of the Gulf Stream for initial dispersal, and the Sargasso Sea as a sea turtle nursery area. The western and north central Sargasso Sea may form some of the earliest nursery areas for cheloniid turtles hatching along Florida's east coast.

*Sargassum* mats form a prominent habitat in the Sargasso Sea; for oceanic-stage turtles, *Sargassum* provides structured habitat with a rich food supply [43], predator protection [44], and thermal benefits promoting growth and feeding [7]. Similar to loggerheads [7], oceanic-stage green turtles in this study remained at the oceanic surface layer and probably received thermal benefits from exposure to direct sunlight at the sea surface. Life in these highly productive, sea surface nursery areas provides the combination of available food and protection with localized warming that can influence temperature-dependent processes in reptiles including digestion and growth [45,46]. Green turtles are known to associate with *Sargassum* habitats during their first years at sea (e.g. [15,16]). Chance sightings of hatchling green turtles in *Sargassum* off the Nicaraguan coast several decades ago provided initial evidence for the importance of *Sargassum* as a nursery habitat for green turtles [24,25]. Smith & Salmon [25]

experimentally documented habitat selection in laboratory and field trials, showing that hatchling green turtles selectively use and burrow into *Sargassum*. Witherington *et al.* [18] documented post-hatchling and small juvenile green turtles using *Sargassum* in the Gulf of Mexico and along Florida's Atlantic coast. Thus, the *Sargassum* habitat is important for young sea turtles during their first year(s) at sea. Presently, the amount of time spent by individual oceanic-stage turtles in this habitat is unclear, and there are yet no empirical studies to determine if species differ in *Sargassum* habitat use and association.

Our empirical track data for both green turtles and loggerheads throw into question the absolute assumptions of hypothesis 2. Instead, we observed differences in behaviour between these species in their use of the Gulf Stream current. Green turtles (15 of 21) in this study departed from the Gulf Stream and the currents of the NASG in greater proportion

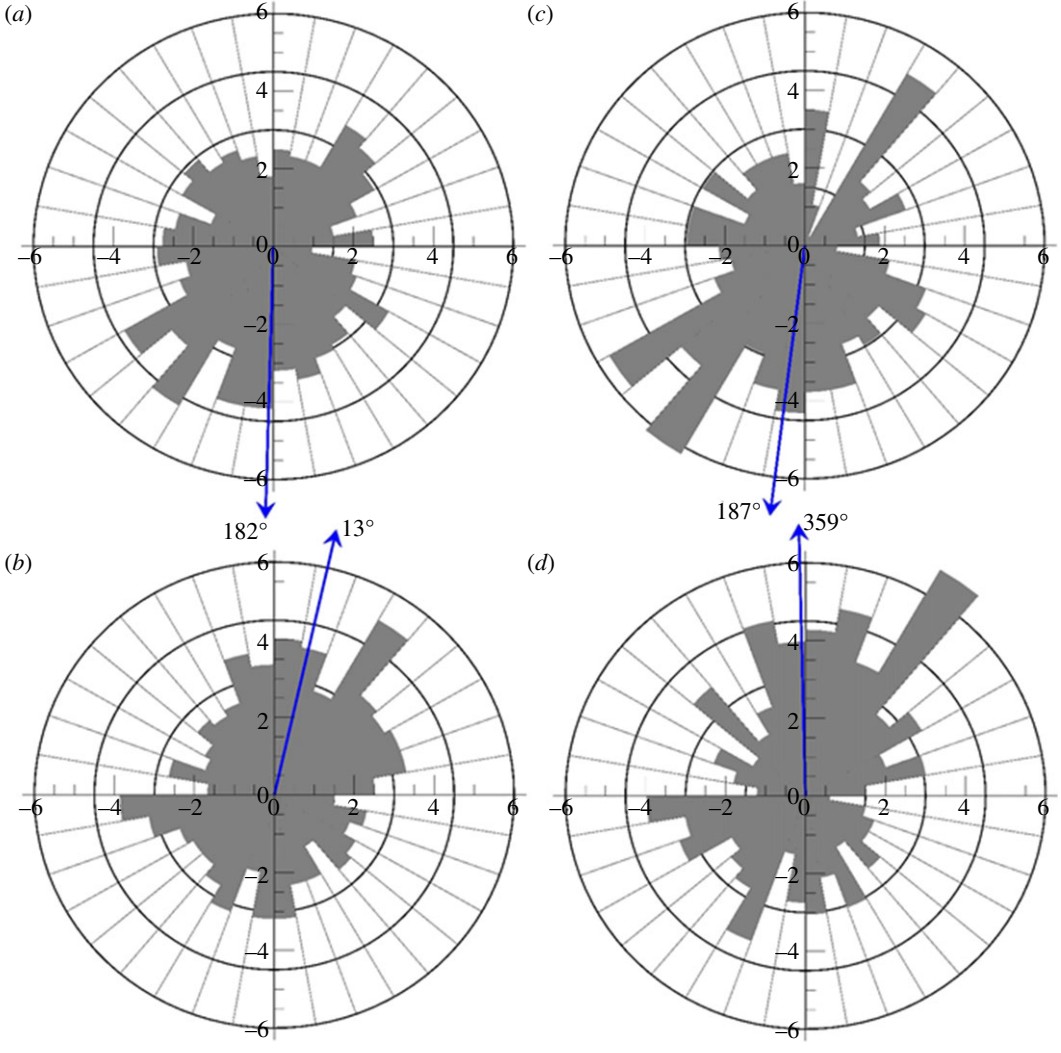

**Figure 4.** (*a*–*d*) Rose diagram frequency distributions of daily turtle orientation comparisons between all pooled green turtles (*a*) and loggerheads (*b*) tracked in the same region, and between 10 green turtles with significant headings (*c*) and 12 loggerheads with significant headings (*d*). The orientations are binned in 10° intervals; the circles show the 1.5, 3.0, 4.5 and 6.0% frequencies. The blue arrow lines indicate the mean orientation. (Online version in colour.)

than loggerheads (7 of 17) observed by Mansfield *et al.* [7]; however, both species originate from the same natal beaches and were released in similar sites and across similar time periods in the Gulf Stream [7]. Almost half of the green turtles in this study departed the Gulf Stream to waters east and south of Cape Hatteras, North Carolina (USA). The remaining animals either entered the Sargasso Sea from more northern positions or were still in the NSTG when their tags ceased—following routes that were similar to those of loggerheads. One question that remains is whether size or age influences if a turtle departs the ocean currents into the Sargasso Sea (are smaller, weaker swimmers more likely to remain entrained in the currents?). Future research should include a larger sample size of turtles representing a variety of sizes and ages in order to examine whether ontogenetic shifts in swimming (and orientation) behaviour occur as turtles grow. Alternatively, as noted by Mansfield *et al.* [7], remotely sensed *Sargassum* biomass data indicate that this floating habitat seasonally distributes north along the eastern US Atlantic coast to the northwestern Atlantic before transitioning to the south and into the Sargasso Sea (for which it is named) [47]. If a turtle associates with floating mats of *Sargassum*, a habitat that provides protection, food, and a thermal benefit, why leave this habitat? Why not continue on with the *Sargassum* into the Sargasso Sea? The turtles

tracked in this study and the loggerheads in our prior work [7] follow a similar route as *Sargassum* biomass in the western North Atlantic. Future research should focus on the behaviour of oceanic-stage sea turtles in association with the 'behaviour' of *Sargassum*—do green turtles actively orient to the Sargasso Sea (thus leave the Gulf Stream current sooner than loggerheads) and do some loggerheads opportunistically ride *Sargassum* mats that might transport them to the Sargasso Sea, while other loggerheads continue on in the NASG? It is possible that annual variation in the availability of *Sargassum* biomass may differ for turtles in this study compared to the earlier study with loggerheads, thereby contributing to some of the differences we observed between the species.

While loggerheads undergo an ontogenetic shift in habitat use and foraging ecology from epipelagic (surface) oceanic to benthic (bottom) neritic habitats at a minimum size of approximately 45 cm SCL [48], juvenile green turtles return to coastal developmental habitats at a much smaller size: 20–30 cm SCL [10]. This suggests that either (i) both species undergo an ontogenetic shift at approximately the same age but have differing growth rates (e.g. loggerheads grow faster than green turtles); (ii) the two species encounter different foraging opportunities and thermal conditions offshore contributing to different growth rates; or (iii) green

turtles spend less time in the oceanic phase of their development. Here, we pose a new question: are green turtles in the North Atlantic more likely to travel shorter distances from their natal origins and remain in the Sargasso Sea as compared to loggerheads that may range farther and/or remain in oceanic habitats longer?

As noted above, the green turtles in this study used and departed from the NASG differently from satellite-tracked loggerheads of similar ages [7]—more green turtles entered the Sargasso Sea, many of which departed the Gulf Stream current sooner (to the south) than the loggerheads. This behavioural difference suggests that while the two species may share many common developmental habitats (oceanic, *Sargassum*) and encounter similar oceanographic features (currents, mesoscale eddies and convergence zones), green turtles may selectively head towards the productive Sargasso Sea waters as demonstrated by their net southern orientation in this study versus the northern/northeastern orientation of loggerheads [7,17,23]. Future work should closely examine the role of *Sargassum* (e.g. size and movements of *Sargassum* mats through time and space), relative to the movements and behaviour of these two species and test whether green turtles are indeed actively orienting to the Sargasso Sea and whether loggerheads may opportunistically end up there if they are simply travelling within *Sargassum* biomass.

## (c) Management implications and conclusion

As more data become available for the sea turtle 'lost years' in the North Atlantic, it is clear that the Sargasso Sea is emerging as an important developmental habitat and nursery for sea turtles. Within US Atlantic waters, Critical Habitat for the loggerhead sea turtle was designated under the Endangered Species Act (ESA) [49]. This designation includes the *Sargassum* habitat essential to the loggerhead sea turtle oceanic life stage and represents the largest spatial designation of Critical Habitat under the ESA. This designation includes regions in the Gulf of Mexico and along the Atlantic US east coast through to the US Exclusive Economic Zone out to 200 nautical miles from shore (which includes part of the western Sargasso Sea). However, critical habitat is not established for green turtles within US Atlantic waters and, importantly, green and loggerhead turtles originating from US rookeries do not remain exclusively within US waters throughout their long lives. Our work highlights the importance of the high seas in the early developmental life stages of sea turtles, and we empirically show that oceanic-stage green turtles from Florida's nesting beaches enter into the shared nursery habitat of the NASG and Sargasso Sea. We encourage the future study of this region to better understand its role in the early life history of Atlantic sea turtles.

Finally, loggerhead and green sea turtles in the North Atlantic venture into the high seas as part of their respective life histories; however, we caution against applying long-held and broad hypotheses (or assumptions) to all species from all rookeries in all ocean basins. Laboratory-reared loggerheads tracked in the South Atlantic oriented differently based on seasonal changes in the available currents offshore of their natal beaches [8]. These turtles also never entered into the interior of the South Atlantic Subtropical Gyre (the southern equivalent to the NASG) similar to their North Atlantic counterparts; some even travelled out of the South Atlantic and north across the Equator into the North Atlantic and Caribbean Sea. Putman & Mansfield [9] demonstrated that green and Kemp's ridley (*Lepidochelys kempii*) turtles in the Gulf of Mexico are active swimmers and exhibit species-specific differences in orientation, challenging the long-held assumption that oceanic-stage turtles are 100% passive drifters. As new technology allows for more turtles of various sizes and ages to be tracked during their first years at sea, we predict that one or even a handful of hypotheses will not fit all.

**Ethics.** Use of animals in research: all research was completed in full compliance with protected species laws and guidelines of the United States and State of Florida, specifically: FAU IACUC protocol A08–40; Florida Marine Turtle Permit MTP-073; and US Fish and Wildlife Service Permits USFWC-TE05127–2.

**Data accessibility.** The data are available from the Dryad Digital Repository: https://doi.org/10.5061/dryad.x95x69ph9 [33].

**Authors' contributions.** K.L.M. and J.W.: designed the study, developed the tag attachment protocols, tagged and released the turtles. J.W.: raised the turtles prior to release. K.L.M.: managed the telemetry data, and J.L.: provided the statistical analyses. All authors contributed to the writing and revision of this manuscript.

All authors gave final approval for publication and agreed to be held accountable for the work performed therein.

**Competing interests.** We declare we have no competing interests.

**Funding.** Support for K.L.M. was provided by the National Academies Research Associateship Program and NOAA Fisheries, Southeast Fisheries Science Center. This study was supported in part by the Florida Sea Turtle License Plate Grants Program to K.L.M., Disney Wildlife Conservation Fund, Save Our Seas Foundation, the Nelligan Sea Turtle Research Support Fund and the Ashwanden Family Fund to J.W.

**Acknowledgements.** We thank S. Epperly, L. Bachler, N. Warriach, M. Young, J. Abernethy, Jim Abernethy's Scuba-Adventures, K. Rusenko, E. Wallace, DDS, R. Salazar DMD and N. Tempel. This work would not have been possible without the help of the Gumbo Limbo Nature Center's sea turtle specialists and the many dedicated FAU sea turtle laboratory students. M. Salmon improved an earlier version of this manuscript and we thank A. Gaos and our referees for their thoughtful reviews. K.L.M. and J.L. dedicate this paper to the memory of Dr Jack Musick, our mentor and friend.

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
