## [Peer Review File · Proceedings of the Royal Society B: Biological Sciences]

Review History

RSPB-2021-0057.R0 (Original submission)

Review form: Reviewer 1

Recommendation

Major revision is needed (please make suggestions in comments)

Scientific importance: Is the manuscript an original and important contribution to its field?

Good

General interest: Is the paper of sufficient general interest?

Good

Quality of the paper: Is the overall quality of the paper suitable?

Good

Is the length of the paper justified?

Yes

Should the paper be seen by a specialist statistical reviewer?

No

Do you have any concerns about statistical analyses in this paper? If so, please specify them explicitly in your report.

No

It is a condition of publication that authors make their supporting data, code and materials available - either as supplementary material or hosted in an external repository. Please rate, if applicable, the supporting data on the following criteria.

Is it accessible?

Yes

Is it clear?

Yes

Is it adequate?

Yes

Do you have any ethical concerns with this paper?

No

Comments to the Author

This study provides the first long-term satellite tracks for neonate green turtles in the western Atlantic Ocean. This is an important study that describes a life stage of a threatened species that remains poorly understood. It therefore brings new and unique information that may help conserve this species. I enjoyed reading the manuscript and seeing these amazing tracks. It is, however, in its current form, mainly a descriptive study with no real hypothesis being tested. It may be possible to modify the manuscript to a more hypothesis-driven study that would fit better into this journal. For instance, answering the following questions might guide the authors: did you expect these results, both in terms of dispersion and comparison with loggerheads? What were your predictions and why? What did you want to test by comparing these tracks to previous published tracks? Does age/size at release make a difference in terms of dispersal pattern?

I have listed below all my comments.

Introduction

Line 28: please replace "is" by 'are'

Line 36: remove "of"

Line 38 : remove "among"

Line 90-94: did these green turtles disperse in the Atlantic Ocean? Does that change your claim that your "data provide the first long-term offshore tracks of oceanic green turtles in the Atlantic Ocean"? It would be good to highlight more what is new in this study compared to the Putman and Mansfield 2015 study

Line 102: Should the current study's results also be compared with ref 21?

Methods

Line 130-132 : repeat of lines 127-128. Please merge

Line 149: are the years correct??

Line 168-169: "shared similar movement patters". Please be more specific here. What is the hypothesis?

Line 170: How was this subset chosen and why? This is important as currently, this seems very arbitrary and not well justified

Line 175-177: long sentence. Please re-phrase

Line 175: please add "green turtle" before "tracks"

Line 179: Please delete : "and the green turtle track data from this study", as this is a repeat of the

previous sentence and confuses the reader.

Results:

Line 193-194: What does this mean? Does this mean that the Eastern Gulf Stream did not act as a barrier? I am not sure any assumptions can be made about that as no turtles reached the Eastern side of the Gulf Stream (i.e. towards Europe). Unless your definition of the eastern side of the Gulf Stream is different from mine?

Line 194-199: 14 turtles entered the Sargasso Sea. What about the 15th turtle? In the next sentence, it is stated that 16 turtles departed the Gulf Stream not 14, please check numbers. If only 3 tagged turtles did not enter the Sargasso Sea out of 21, 18 should have entered. In the previous sentences, it is stated 15 or 16 entered the Sargasso Sea. Please clarify.

Is it also possible to test if the turtles that did not enter the Sargasso Sea were smaller and/or younger than the ones that did? Maybe smaller turtles were not as strong swimmers? Maybe age/size at release make a difference in terms of dispersal pattern?

Line 2020: replace "codes" by 'classes'

Line 203: Did the tracked turtles remain at the sea surface all day long? Is there actually a way to know? If not, maybe add "mostly"

Line 204: please add "and the rest were locations classes A and B"

Line 213-217: Is it possible to show the Sargasso Sea and maybe the Gulf Stream on Figure 3?

Line 215 and 217 : "to the east of the Gulf Stream boundary". Do you mean to the west? For me, the east of the gulf stream is towards Europe, or do you mean the eastern boundary of the southwest part of the Gulf Stream?

Line 220: please remove "were"

Line 22: Please explain how and why these 17 individuals were selected for this analysis? Why were the 10 individuals with statistically significant orientation not selected instead? I am not entirely comfortable with pooling individuals together while some of them had significant orientation and other didn't. Please justify.

Line 227-228: please replace 'is' by "was"

Discussion

Line 239 : remove "of"

Line 286-287: Were the turtles release at the same time of the year? If not, could that have had implications for the differences between the two species?

Line 296-299: maybe modify as : "This suggests that either (a) both species undergo an ontogenetic shift at the same age and therefore loggerheads grow faster than green turtles or they don't and (b) green turtles spend less time in the oceanic phase of their development"

Line 299-302: this was not explicitly tested in this study. Please either remove from discussion or add corresponding results.

Line 304-311: This is an interesting finding. Any hypothesis as to why there is such a difference between these two species? This paragraph seems to indicate that the young loggerheads use a different part of the Sargasso Sea, is that correct? Or would you still say that they share the same nursery (as suggested in the conclusion). Maybe be more specific in what is similar between the two species and what is different and why? And if these differences are meaningful in terms of conservation or if we simply need to protect a larger area to encompass both nursery habitats?

Table 1: please define "body depth". The term 'static water' is used in the legend but not in the main results. Please make sure oceanographic descriptions are consistent throughout the manuscript.

Line 403: please change "to enter" by "entering"

Figure 4 , Line 517: I thought the orientation headings were calculated every 6 hours and not daily. Please check with methods and modify if necessary.

Table 2.a.: the U2 score for the first turtle is missing.

Review form: Reviewer 2 (Alexander Gaos)

Recommendation

Accept with minor revision (please list in comments)

Scientific importance: Is the manuscript an original and important contribution to its field?

Excellent

General interest: Is the paper of sufficient general interest?

Excellent

Quality of the paper: Is the overall quality of the paper suitable?

Good

Is the length of the paper justified?

Yes

Should the paper be seen by a specialist statistical reviewer?

No

Do you have any concerns about statistical analyses in this paper? If so, please specify them explicitly in your report.

Yes

It is a condition of publication that authors make their supporting data, code and materials available - either as supplementary material or hosted in an external repository. Please rate, if applicable, the supporting data on the following criteria.

Is it accessible?

N/A

Is it clear?

N/A

Is it adequate?

N/A

Do you have any ethical concerns with this paper?

No

Comments to the Author

This study focuses on satellite tracking of young green turtles tagged off the eastern coast of the USA. It is generally well written and presents novel movement information for a life-stage that remains largely unknown for most sea turtle species and populations, thus represents a wonderful contribution that I will be excited to see in print.

Notwithstanding this perspective, I have several suggestions that I think would improve the manuscript, the most important of which relate to the figures:

- Include outlines or some type of shading to show the general areas of the 1) Gulf Stream and 2) Sargasso Sea. These areas are discussed throughout the manuscript, are described as discrete areas, and analyses are conducted on locations within them, but they are not depicted anywhere. Would be a huge improvement to provide a visual of their general location/boundaries.

- Zoom in on the tracks presented in Figure 2 and Figure 3 so more detail can be seen on individual tracks. Particularly relevant when citing particular tracks (e.g., lines 190-193) If authors would like to show range of tracks within Atlantic, I suggest using an inset to do so (i.e., zoom main image in on tracks, then place a small inset in the one of the corners depicting tracks within entire Atlantic).
- Lines 213-215 – I agree that the kernel analyses are greatly biased by their initial release location and the Gulf Stream. I think doing an analysis for only the locations outside the Gulf Stream would provide a better visual for the importance of the Sargasso Sea. Suggest authors try this.
- Adding labels to the actual maps in Figure 3 that indicate the species included in each pane would facilitate interpretation.
- Adding labels to the rose diagrams in figure 4 would also facilitate interpretation. However, this figure is also confusing. Line 219-221 discussed data from 10 and 11 turtles for which orientation analysis was significant and non-significant, respectively, then references Fig. 4a and b. However, Fig. 4a and b are for individual “example” turtles (at least that’s my understanding from reading the figure description), so this doesn’t seem appropriate. Why showing “example” turtles for Fig. 4a, b, d, and e? Why not showing combined results for the groups of green and loggerhead turtles? Could be misinterpreted as cherry picking the results. Suggest clarifying this figure and how it corresponds to the text.
- Because this study really aims to compare and contrast green and loggerhead tracks, would recommend a map that shows the tracks for both species, I would also consider including a map that contains tracks from both species/studies.

The manuscript states that all sea turtle species (except flatbacks) migrate from natal neritic waters to oceanic nursery areas, then recruit to neritic area at a certain size (Line 50-52 & 324-326). Although I would agree that some (most?) populations of all species may undergo this movement pattern, I’d argue that the jury is still out on whether all populations of all species do. For instance, tracking research of young-of-year hawksbill turtles that I am involved with (currently in the process of being written up by Dr. Mike Liles) has found that many of the turtles migrate directly into the estuarine waters adjacent to the beaches where they were born. By stating that offshore migrations is a given, I think the authors actually undercut the novelty of their study (and the previous Mansfield manuscript), which are really the first to demonstrate this actually occurs (rather than simply being a hypothesis) and provides justification for why much more of this type of research is needed. I recommend leaving this as behavior that has been confirmed for the two species/populations in question, but remains an open question for many other populations.

I recommend the authors consider rewriting much of the first paragraph in the discussion (specifically lines 237-248). The authors start by contrasting the initial beaches/habitats of green and loggerhead turtles in the study areas. I was expecting this to transition into how they tend to use different oceanic areas for development, i.e., lines 308-310 seem to be the more relevant take-home message and I think it would be more effective to make this point (condensed version) at the start of the introduction (i.e., could still leave lines 308-310 as they are). However, rather than doing that, the first paragraph goes into the morphology of the two species and discusses the different tag adhesion methods, which would seem more appropriate for a “methods” manuscript. I recommend restructuring the opening discussion paragraph. Consider a separate section somewhere else discussing tag adherence, longevity, etc., or omit that part altogether.

Additional comments:

Line 29 – this assumes sea turtles undertake movements over 1,000’s of kilometers across open ocean. This isn’t necessarily always the case and certainly hasn’t been shown for many species or populations (hence the novelty of this study).

Line 30 – “Long-term” and “very young”, the former used in several parts of the manuscript, are quite ambiguous terms. I would try to be more specific (or define what you consider “long-term” and why).

Line 36 – Remove “of”

Line 48 – There are multiple definitions for the term “oceanic” so it might be worth defining it here as relating to the open-ocean just to be clear. Line 81 says, “(remaining off of the Continental Shelf or in waters >200 m depth)”...is that what the authors consider oceanic?

Line 53 – what is the “L.” after *Chelonia mydas*?

Line 68 – Lines 253-254 in the discussion section define the Sargasso Sea. I suggest moving that description here.

Line 68 – define ENE.

Line 72-74 – Suggest providing an example(s), or at least a citation(s).

Line 78 – “Turtles in this study” is a bit of an unclear reference. I initially thought the authors were referring to turtles in the current study and it threw me off. Suggest tweaking to make clear.

Line 80 and 251 – “long standing hypotheses” by who? Provide a reference(s).

Line 96 – Given Mansfield is the first author of this manuscript, I would suggest rewording “Building on the work of Mansfield et al. [19, 20, 22]...” to “Building on previous studies (Mansfield et al. [19, 20, 22])...”

Line 133 – what constitutes “adapted well”?

Line 161 – Missing closing parenthesis after citations.

Line 170-171 – Describe how/why was this geographic subset chosen. I know it was done to test differences in mean orientation, but explain why tracks within this geographic area.

Line 190-193 – I cannot distinguish the referenced tracks in Figure 2 (see my previous comment regarding zooming in for this figure).

Line 194-199 – These numbers don’t seem to coincide/add up. First says 15 of the 21 turtles departed the Gulf Stream, then says eight departed south of Cape Hatteras and eight departed north/east. Also, $8+8+3$ (that had not entered the Sargasso Sea prior to tag cessation) = 19, so what about the other two turtles. Please make this clearer.

Line 202 – It would seem prudent to mention something in the methods about how tag charging data is collected.

Line 203 – reads as if tracked turtles ALWAYS remained at the surface, but cannot be sure.

Line 205 – suggest moving to methods

Line 206 – Add “V” after 4.04.

Line 285 – provide the proportion (%) of loggerheads that departed the Gulf Stream in the previous manuscript.

Line 304-305 – seems that this sentence should proceed the previous paragraph.

Lines 323-324 – the purpose of including the portion of the sentence, “no sea turtle species remains exclusively within US waters throughout their long lives” is unclear to me. Elaborate.

Congratulations on a great manuscript and I hope my comments are useful.

Kind regards,
Alexander Gaos

Decision letter (RSPB-2021-0057.R0)

13-Mar-2021

Dear Dr Mansfield:

Your manuscript has now been peer reviewed and the reviews have been assessed by an Associate Editor. The reviewers' comments (not including confidential comments to the Editor) and the comments from the Associate Editor are included at the end of this email for your reference. As you will see, the reviewers and the Editors have raised some concerns with your manuscript and we would like to invite you to revise your manuscript to address them.

Research ethics:

Use of animals and field studies:

It is a condition of publication that you make available the data and research materials supporting the results in the article. Please see our Data Sharing Policies (<https://royalsociety.org/journals/authors/author-guidelines/#data>). Datasets should be deposited in an appropriate publicly available repository and details of the associated accession number, link or DOI to the datasets must be included in the Data Accessibility section of the article (<https://royalsociety.org/journals/ethics-policies/data-sharing-mining/>). Reference(s) to datasets should also be included in the reference list of the article with DOIs (where available).

Please submit a copy of your revised paper within three weeks. If we do not hear from you within this time your manuscript will be rejected. If you are unable to meet this deadline please let us know as soon as possible, as we may be able to grant a short extension.

Best wishes,
Dr Daniel Costa
<mailto:proceedingsb@royalsociety.org>

Associate Editor

Board Member: 1

Comments to Author:

Two expert reviewers have assessed the manuscript and both agree it presents some exciting and novel information, extending the previous work in loggerheads and revealing some important species differences in movement patterns and distance traveled that allows consideration of the larger critical habitat use by the taxon. However, the first reviewer has hit on an important potential deficit of the paper as written for the Proc B readership, in that the manuscript is written as mostly descriptive and fails to generate any hypothesis driven basis for expectations of differences between these species. I think the reviewer is correct that this would be a much stronger effort if the authors could cast the study in terms of testing hypotheses about the species differences and has provided the authors with a number of suggestions for doing that. This should be possible for the authors as they have published previous work based on predicting the movement patterns of oceanic turtles using life history traits, particle simulations, oceanic circulation and movement ecology theory. While the results presented are exciting, this approach would help raise the level of the paper and increase the interest and suitability of the paper for Proc B. Both reviewers provided some important detailed comments towards revision. I would like to ask the authors to revise the paper around the broad comments of the first reviewer and detailed specific comments of both reviewers.

Reviewer(s)' Comments to Author:

Referee: 1

Comments to the Author(s)

This study provides the first long-term satellite tracks for neonate green turtles in the western Atlantic Ocean. This is an important study that describes a life stage of a threatened species that remains poorly understood. It therefore brings new and unique information that may help conserve this species. I enjoyed reading the manuscript and seeing these amazing tracks. It is, however, in its current form, mainly a descriptive study with no real hypothesis being tested. It may be possible to modify the manuscript to a more hypothesis-driven study that would fit better into this journal. For instance, answering the following questions might guide the authors: did you expect these results, both in terms of dispersion and comparison with loggerheads? What were your predictions and why? What did you want to test by comparing these tracks to previous published tracks? Does age/size at release make a difference in terms of dispersal pattern?

I have listed below all my comments.

Introduction

Line 28: please replace "is" by 'are'

Line 36: remove "of"

Line 38 : remove "among"

Line 90-94: did these green turtles disperse in the Atlantic Ocean? Does that change your claim that your "data provide the first long-term offshore tracks of oceanic green turtles in the Atlantic Ocean"? It would be good to highlight more what is new in this study compared to the Putman and Mansfield 2015 study

Line 102: Should the current study's results also be compared with ref 21?

Methods

Line 130-132 : repeat of lines 127-128. Please merge

Line 149: are the years correct??

Line 168-169: "shared similar movement patters". Please be more specific here. What is the hypothesis?

Line 170: How was this subset chosen and why? This is important as currently, this seems very arbitrary and not well justified

Line 175-177: long sentence. Please re-phrase

Line 175: please add "green turtle" before "tracks"

Line 179: Please delete : “and the green turtle track data from this study”, as this is a repeat of the previous sentence and confuses the reader.

Results:

Line 193-194: What does this mean? Does this mean that the Eastern Gulf Stream did not act as a barrier? I am not sure any assumptions can be made about that as no turtles reached the Eastern side of the Gulf Stream (i.e. towards Europe). Unless your definition of the eastern side of the Gulf Stream is different from mine?

Line 194-199: 14 turtles entered the Sargasso Sea. What about the 15th turtle? In the next sentence, it is stated that 16 turtles departed the Gulf Stream not 14, please check numbers. If only 3 tagged turtles did not enter the Sargasso Sea out of 21, 18 should have entered. In the previous sentences, it is stated 15 or 16 entered the Sargasso Sea. Please clarify.

Is it also possible to test if the turtles that did not enter the Sargasso Sea were smaller and/or younger than the ones that did? Maybe smaller turtles were not as strong swimmers? Maybe age/size at release make a difference in terms of dispersal pattern?

Line 2020: replace “codes” by ‘classes’

Line 203: Did the tracked turtles remain at the sea surface all day long? Is there actually a way to know? If not, maybe add “mostly”

Line 204: please add “and the rest were locations classes A and B”

Line 213-217: Is it possible to show the Sargasso Sea and maybe the Gulf Stream on Figure 3?

Line 215 and 217 : “to the east of the Gulf Stream boundary”. Do you mean to the west? For me, the east of the gulf stream is towards Europe, or do you mean the eastern boundary of the southwest part of the Gulf Stream?

Line 220: please remove “were”

Line 22: Please explain how and why these 17 individuals were selected for this analysis? Why were the 10 individuals with statistically significant orientation not selected instead? I am not entirely comfortable with pooling individuals together while some of them had significant orientation and other didn't. Please justify.

Line 227-228: please replace ‘is’ by “was”

Discussion

Line 239 : remove “of”

Line 286-287: Were the turtles released at the same time of the year? If not, could that have had implications for the differences between the two species?

Line 296-299: maybe modify as : “This suggests that either (a) both species undergo an ontogenetic shift at the same age and therefore loggerheads grow faster than green turtles or they don't and (b) green turtles spend less time in the oceanic phase of their development”

Line 299-302: this was not explicitly tested in this study. Please either remove from discussion or add corresponding results.

Line 304-311: This is an interesting finding. Any hypothesis as to why there is such a difference between these two species? This paragraph seems to indicate that the young loggerheads use a different part of the Sargasso Sea, is that correct? Or would you still say that they share the same nursery (as suggested in the conclusion). Maybe be more specific in what is similar between the two species and what is different and why? And if these differences are meaningful in terms of conservation or if we simply need to protect a larger area to encompass both nursery habitats?

Table 1: please define “body depth”. The term ‘static water’ is used in the legend but not in the main results. Please make sure oceanographic descriptions are consistent throughout the manuscript.

Line 403: please change “to enter” by “entering”

Figure 4 , Line 517: I thought the orientation headings were calculated every 6 hours and not daily. Please check with methods and modify if necessary.

Table 2.a.: the U2 score for the first turtle is missing.

Referee: 2

Comments to the Author(s)

This study focuses on satellite tracking of young green turtles tagged off the eastern coast of the USA. It is generally well written and presents novel movement information for a life-stage that remains largely unknown for most sea turtle species and populations, thus represents a wonderful contribution that I will be excited to see in print.

Notwithstanding this perspective, I have several suggestions that I think would improve the manuscript, the most important of which relate to the figures:

- Include outlines or some type of shading to show the general areas of the 1) Gulf Stream and 2) Sargasso Sea. These areas are discussed throughout the manuscript, are described as discrete areas, and analyses are conducted on locations within them, but they are not depicted anywhere. Would be a huge improvement to provide a visual of their general location/boundaries.
- Zoom in on the tracks presented in Figure 2 and Figure 3 so more detail can be seen on individual tracks. Particularly relevant when citing particular tracks (e.g., lines 190-193) If authors would like to show range of tracks within Atlantic, I suggest using an inset to do so (i.e., zoom main image in on tracks, then place a small inset in the one of the corners depicting tracks within entire Atlantic).
- Lines 213-215 – I agree that the kernel analyses are greatly biased by their initial release location and the Gulf Stream. I think doing an analysis for only the locations outside the Gulf Stream would provide a better visual for the importance of the Sargasso Sea. Suggest authors try this.
- Adding labels to the actual maps in Figure 3 that indicate the species included in each pane would facilitate interpretation.
- Adding labels to the rose diagrams in figure 4 would also facilitate interpretation. However, this figure is also confusing. Line 219-221 discussed data from 10 and 11 turtles for which orientation analysis was significant and non-significant, respectively, then references Fig. 4a and b. However, Fig. 4a and b are for individual “example” turtles (at least that’s my understanding from reading the figure description), so this doesn’t seem appropriate. Why showing “example” turtles for Fig. 4a, b, d, and e? Why not showing combined results for the groups of green and loggerhead turtles? Could be misinterpreted as cherry picking the results. Suggest clarifying this figure and how it corresponds to the text.
- Because this study really aims to compare and contrast green and loggerhead tracks, would recommend a map that shows the tracks for both species, I would also consider including a map that contains tracks from both species/studies.

The manuscript states that all sea turtle species (except flatbacks) migrate from natal neritic waters to oceanic nursery areas, then recruit to neritic area at a certain size (Line 50-52 & 324-326). Although I would agree that some (most?) populations of all species may undergo this movement pattern, I’d argue that the jury is still out on whether all populations of all species do. For instance, tracking research of young-of-year hawksbill turtles that I am involved with (currently in the process of being written up by Dr. Mike Liles) has found that many of the turtles migrate directly into the estuarine waters adjacent to the beaches where they were born. By stating that offshore migrations is a given, I think the authors actually undercut the novelty of their study (and the previous Mansfield manuscript), which are really the first to demonstrate this actually occurs (rather than simply being a hypothesis) and provides justification for why much more of this type of research is needed. I recommend leaving this as behavior that has been confirmed for the two species/populations in question, but remains an open question for many other populations.

I recommend the authors consider rewriting much of the first paragraph in the discussion (specifically lines 237-248). The authors start by contrasting the initial beaches/habitats of green and loggerhead turtles in the study areas. I was expecting this to transition into how they tend to use different oceanic areas for development, i.e., lines 308-310 seem to be the more relevant take-home message and I think it would be more effective to make this point (condensed version) at the start of the introduction (i.e., could still leave lines 308-310 as they are). However, rather than doing that, the first paragraph goes into the morphology of the two species and discusses the different tag adhesion methods, which would seem more appropriate for a "methods" manuscript. I recommend restructuring the opening discussion paragraph. Consider a separate section somewhere else discussing tag adherence, longevity, etc., or omit that part altogether.

Additional comments:

Line 29 - this assumes sea turtles undertake movements over 1,000's of kilometers across open ocean. This isn't necessarily always the case and certainly hasn't been shown for many species or populations (hence the novelty of this study).

Line 30 - "Long-term" and "very young", the former used in several parts of the manuscript, are quite ambiguous terms. I would try to be more specific (or define what you consider "long-term" and why).

Line 36 - Remove "of"

Line 48 - There are multiple definitions for the term "oceanic" so it might be worth defining it here as relating to the open-ocean just to be clear. Line 81 says, "(remaining off of the Continental Shelf or in waters >200 m depth)"...is that what the authors consider oceanic?

Line 53 - what is the "L." after *Chelonia mydas*?

Line 68 - Lines 253-254 in the discussion section define the Sargasso Sea. I suggest moving that description here.

Line 68 - define ENE.

Line 72-74 - Suggest providing an example(s), or at least a citation(s).

Line 78 - "Turtles in this study" is a bit of an unclear reference. I initially thought the authors were referring to turtles in the current study and it threw me off. Suggest tweaking to make clear.

Line 80 and 251 - "long standing hypotheses" by who? Provide a reference(s).

Line 96 - Given Mansfield is the first author of this manuscript, I would suggest rewording "Building on the work of Mansfield et al. [19, 20, 22]..." to "Building on previous studies (Mansfield et al. [19, 20, 22])..."

Line 133 - what constitutes "adapted well"?

Line 161 - Missing closing parenthesis after citations.

Line 170-171 - Describe how/why was this geographic subset chosen. I know it was done to test differences in mean orientation, but explain why tracks within this geographic area.

Line 190-193 - I cannot distinguish the referenced tracks in Figure 2 (see my previous comment regarding zooming in for this figure).

Line 194-199 - These numbers don't seem to coincide/add up. First says 15 of the 21 turtles departed the Gulf Stream, then says eight departed south of Cape Hatteras and eight departed

north/east. Also, $8+8+3$ (that had not entered the Sargasso Sea prior to tag cessation) = 19, so what about the other two turtles. Please make this clearer.

Line 202 – It would seem prudent to mention something in the methods about how tag charging data is collected.

Line 203 – reads as if tracked turtles ALWAYS remained at the surface, but cannot be sure.

Line 205 – suggest moving to methods

Line 206 – Add “V” after 4.04.

Line 285 – provide the proportion (%) of loggerheads that departed the Gulf Stream in the previous manuscript.

Line 304-305 – seems that this sentence should proceed the previous paragraph.

Lines 323-324 – the purpose of including the portion of the sentence, “no sea turtle species remains exclusively within US waters throughout their long lives” is unclear to me. Elaborate.

Congratulations on a great manuscript and I hope my comments are useful.

Kind regards,

Alexander Gaos

Author's Response to Decision Letter for (RSPB-2021-0057.R0)

See Appendix A.

Decision letter (RSPB-2021-0057.R1)

12-Apr-2021

Dear Dr Mansfield

I am pleased to inform you that your manuscript entitled "First Atlantic Satellite Tracks of “Lost Years” Green Turtles Support the Importance of the Sargasso Sea as a Sea Turtle Nursery" has been accepted for publication in Proceedings B.

Data Accessibility section

Open Access

Paper charges

Sincerely,

Dr Daniel Costa

Appendix A

Proceedings of the Royal Society B – Manuscript ID: RSPB-2021-0057

Dear Dr. Costa, Associate Editor:

We are pleased to resubmit our edited manuscript titled: *First Atlantic Satellite Tracks of “Lost Years” Green Turtles Support the Importance of a Sargasso Sea as a Sea Turtle Nursery* (PRSB-2021-0057) for publication consideration in the *Proceedings of the Royal Society B*. All authors contributed to the revisions and referee response. We’ve addressed all referee comments and concerns and our responses to the reviews are compiled below. We have also uploaded the revised track-changed manuscript (and a clean version for ease of reading) and updated figures/tables as appropriate/requested based on reviews.

We thank the referees for their thoughtful reviews and feel that this revised version is a stronger paper as a result. We now model the manuscript after our loggerhead paper (Mansfield et al. 2014) and more directly address long-held assumptions/hypotheses about the sea turtle lost years. We also include new analyses (Figure 3) per Reviewer 2’s request where we include kernel analyses that are focused solely on the Sargasso Sea (we did opt to retain the original analyses with all tracks since we feel that all data will be of interest to conservation managers and the broader sea turtle research community).

As noted with our original submission, we have a number of beautiful images of the young sea turtles when released that we are happy to provide for media purposes if our manuscript is accepted for publication. Thank you for your consideration.

Regards,

Revisions and responses to referees

We thank our reviewers for their thoughtful and helpful reviews. Below, we provide a point-by-point response to their comments/suggestions. Please note that all new line numbers mentioned below refer to the **track-changed** file, not the clean “reading” file (with all track changes accepted). We are excited about our revised manuscript and think that it is a stronger paper as a result of the reviews.

General editorial comments:

Use of animals and field studies: This study includes vertebrate animals and we do specify our relevant protected species permits and IACUC protocols in the Acknowledgement section (see lines 526-529 in the track-changed file).

Data accessibility and data citation: We established a Dryad entry with the track dataset we used in this manuscript at the time of original submission. Our DOI and Dryad citation is:

Mansfield, Katherine; Wyneken, Jeanette; Luo, Jiangang (2021), First Atlantic satellite tracks of “lost years” green turtles support the importance of a Sargasso Sea as a sea turtle nursery, Dryad, Dataset, <https://doi.org/10.5061/dryad.x95x69ph9>

We are updating this Dryad entry to include the track data from our 2014 loggerhead paper since these data are available in a different (non-Dryad) repository and we want to make them available in reference to this paper as well (and with an associated DOI). We now include this citation as relevant in the text and References section and in the new Data accessibility and citation section (starting line 531).

Competing interest statement: added starting line 536.

Author contribution statement: added starting line 539.

Referee Comments:

**Associate Editor
Board Member: 1**

Comments to Author:

Two expert reviewers have assessed the manuscript and both agree it presents some exciting and novel information, extending the previous work in loggerheads and revealing some important species differences in movement patterns and distance traveled that allows consideration of the larger critical habitat use by the taxon. However, the first reviewer has hit on an important potential deficit of the paper as written for the Proc B readership, in that the manuscript is written as mostly descriptive and fails to generate any hypothesis driven basis for expectations of differences between these species. I think the reviewer is correct

that this would be a much stronger effort if the authors could cast the study in terms of testing hypotheses about the species differences and has provided the authors with a number of suggestions for doing that. This should be possible for the authors as they have published previous work based on predicting the movement patterns of oceanic turtles using life history traits, particle simulations, oceanic circulation and movement ecology theory. While the results presented are exciting, this approach would help raise the level of the paper and increase the interest and suitability of the paper for Proc B. Both reviewers provided some important detailed comments towards revision. I would like to ask the authors to revise the paper around the broad comments of the first reviewer and detailed specific comments of both reviewers.

Thank you for these comments. We originally structured the paper in a similar way to our first paper (Mansfield et al. 2014) that presented the long-held assumptions (“hypotheses”) associated with the sea turtle ‘lost years’ as a whole, within the context of loggerheads. A last-minute edit on my part (first author) deleted a similar approach in this paper in an interest of word count. I am happy to include the original text again as it makes the paper comparable to our other work and also helps target and break down the long-held assumptions about the sea turtle ‘lost years’ that our research community has held on to for so long. See additions added throughout the revised Background (in particular lines 65-85, 142-145, 153-161) and references to these hypotheses throughout the Discussion.

Reviewer(s)' Comments to Author:

Referee: 1

Comments to the Author(s)

This study provides the first long-term satellite tracks for neonate green turtles in the western Atlantic Ocean. This is an important study that describes a life stage of a threatened species that remains poorly understood. It therefore brings new and unique information that may help conserve this species. I enjoyed reading the manuscript and seeing these amazing tracks. It is, however, in its current form, mainly a descriptive study with no real hypothesis being tested. It may be possible to modify the manuscript to a more hypothesis-driven study that would fit better into this journal. For instance, answering the following questions might guide the authors: did you expect these results, both in terms of dispersion and comparison with loggerheads? What were your predictions and why? What did you want to test by comparing these tracks to previous published tracks?

Thank you – we include three long-held assumptions/hypotheses about the early oceanic stage of sea turtles in our Background section and specifically note that we test these hypotheses as part of our study objectives (lines 65-85, 142-145, 153-161). The questions the reviewer suggests (above) are now worked in to the narrative of the Background along with specific hypotheses we test.

Does age/size at release make a difference in terms of dispersal pattern?

This is an excellent question; however, our sample size is too small and too discrete in terms of the range of turtles' sizes we tagged to be able to answer this question at this time. All turtles were of a comparable age and size (and all were of a size/age estimated to be found offshore or actually encountered offshore, e.g., Putman and Mansfield 2015). With more turtles tagged in the future (and other studies), we hope to more closely examine ontogenetic behavioral changes across sizes or ages (e.g., what percent of the time are turtles active vs. passive and does this change as they grow/age? What role does *Sargassum* play in this behavior).

I have listed below all my comments.

Introduction

Line 28: please replace “is” by ‘are’

Done (good catch, thank you).

Line 36: remove “of”

Removed.

Line 38 : remove “among”

Removed.

Line 90-94: did these green turtles disperse in the Atlantic Ocean? Does that change your claim that your “data provide the first long-term offshore tracks of oceanic green turtles in the Atlantic Ocean”? It would be good to highlight more what is new in this study compared to the Putman and Mansfield 2015 study

This is an artifact of publication timing, unfortunately. This study was conducted prior to the work published in 2015 but we were slow to complete this manuscript. So, technically these do represent the first green turtle tracks and this is why we clarified line 38 (above) with the word ‘among’. Since I am the PI on both studies, I am familiar with the timelines, but realize the reader may not be. Please see edits within the text that hopefully help clarify (lines 135-138).

The 2015 study applied the techniques from this work to wild-caught turtles in the Gulf of Mexico, specifically tackling a 4th long-held hypothesis about oceanic stage sea turtles: that they are 100% passive drifters. The 2015 project deployed passive oceanographic drifters with all tagged turtles to directly test for passive or active behaviour. Unfortunately, we did not have the opportunity to deploy drifters with the study detailed in this manuscript (the drifter work idea was sparked by this work and our prior loggerhead work per Mansfield et al. 2014). So the 2015 work is complimentary, but doesn't directly tackle the other hypotheses (oceanic, passive drift in the North Atlantic Subtropical Gyre currents, surface-dwelling). Given that we encounter very few loggerheads in our Gulf of Mexico work (n=3 over 10 years of trying to catch turtles in the Gulf), that study also cannot provide comparison to loggerheads in the same way as

this paper does (turtles from the same natal beaches and who would likely encounter the same oceanographic features when transitioning from the nesting beaches offshore). Part of what our work is helping to shed light on is the fact that we can't assume that the old hypotheses can be applied to all turtles from all rookeries in all ocean basins...something we try to drive home with this paper (and with responses/edits addressing Reviewer 2's suggestions).

Line 102: Should the current study's results also be compared with ref 21?

See comment above. We encounter green turtles from very different rookeries in the Gulf of Mexico than the Florida rookeries, and have only captured a few loggerheads there in the 10+ years we've been sampling. The green turtles in the Gulf of Mexico originate from Mexico, the Caribbean, and a few from Florida (Phillips et al. in prep – part of the doctoral work of one of my PhD students). These turtles encounter very different ocean currents and the Continental Shelf is quite different in the Gulf of Mexico (the West Florida Shelf extends into waters that support oceanic habitat such as *Sargassum*). Historically, the hypotheses/assumptions regarding the early life stages of loggerheads and greens were all based on dispersal from nesting beaches along the Atlantic US coast (where ~90% of all North Atlantic loggerheads and ~12-15% of all green turtles nest; 80% of green turtle nesting is in Costa Rica with small numbers originating elsewhere in the Caribbean and Mexico). So, for comparison and to highlight the Sargasso Sea as a potential sea turtle nursery area, the primary comparison in this paper is between green and loggerhead turtles that were lab-reared, who hatched from the same beaches at similar times of year, and that would encounter the same oceanographic conditions – this is important considering that these same turtles likely have innate compass senses and magnetic map sense (per Dr. Ken Lohmann's large body of work on the subject) and we do not know if turtles we encounter in the Gulf of Mexico have the same 'maps'... Bottom line we are finding that the more we tag and track these young turtles that the old hypotheses/assumptions do not hold up everywhere. Something we try to drive home in this paper and will continue to do so in subsequent papers that are forthcoming for the Gulf of Mexico. This paper is one piece of that logical puzzle.

All that said, please see text for some edits/clarification that address this (Background and Discussion sections).

Methods

Line 130-132: repeat of lines 127-128. Please merge
Sentences were edited and merged.

Line 149: are the years correct??

Yes, please refer to Tables 1 and 2 for additional detail. As mentioned above, it has taken us some time to get this manuscript out (many life changes in the interim).

Line 168-169: "shared similar movement patters". Please be more specific here. What is the hypothesis?

Altered the text to better reflect the hypotheses stated in the Background and clarified what it is we specifically were asking/testing.

Line 170: How was this subset chosen and why? This is important as currently, this seems very arbitrary and not well justified

We now provide more detail in the text (lines 249-254). This subset represents the spatial range within which both green and loggerhead tracks occurred, and only a few individual tracks extend beyond this spatial range (the northern limit of the NASG). The subset of locations includes the region tested in loggerheads from Mansfield et al. [2014] encompassing the experimentally-generated magnetic field locations used by Lohmann & Lohmann [2006]—a region where laboratory-reared naive loggerheads oriented to the ENE to theoretically remain within the NASG.

Line 175-177: long sentence. Please re-phrase

Re-phrased.

Line 175: please add “green turtle” before “tracks”

“Green turtle” is now added.

Line 179: Please delete: “and the green turtle track data from this study”, as this is a repeat of the previous sentence and confuses the reader.

Deleted.

Results:

Line 193-194: What does this mean? Does this mean that the Eastern Gulf Stream did not act as a barrier? I am not sure any assumptions can be made about that as no turtles reached the Eastern side of the Gulf Stream (i.e. towards Europe). Unless your definition of the eastern side of the Gulf Stream is different from mine?

This refers to the eastern edge of the Gulf Stream current that is the boundaries to the Sargasso Sea (in the western Atlantic)—imagine the current as a river that generally flows to the NE or ENE along the US Atlantic coast. The river has a west (left) “bank” or frontal boundary and an east “bank” or frontal boundary. Turtles generally did not travel to the west across the west “bank” outside the Gulf Stream current, but they did leave the current from the eastern frontal boundary of the current (east “bank”), traveling into the Sargasso Sea... We agree that this is a confusing sentence and edited the text throughout the manuscript to help clarify, hopefully. See ~lines 274-280.

Line 194-199: 14 turtles entered the Sargasso Sea. What about the 15th turtle? In the next sentence, it is stated that 16 turtles departed the Gulf Stream not 14, please check numbers. If only 3 tagged turtles did not enter the Sargasso Sea out of 21, 18 should have entered. In the previous sentences, it is stated 15 or 16 entered the Sargasso Sea. Please clarify.

Thank you for catching this, there was a typo in this section (an artifact of an earlier edit). We accidentally deleted a sentence about the 15th turtle that traveled onto the Continental Shelf just prior to its tag ceasing transmission (this has been added to the

text). We also accidentally duplicated the number of turtles transitioning to the Sargasso Sea north and south of Cape Hatteras—this is also corrected in the text (and we added the one turtle that traveled to the Continental Shelf to the number leaving the Gulf Stream south of Hatteras). Again, thanks for catching this (see lines 278-284).

Is it also possible to test if the turtles that did not enter the Sargasso Sea were smaller and/or younger than the ones that did? Maybe smaller turtles were not as strong swimmers? Maybe age/size at release make a difference in terms of dispersal pattern?

This is an excellent suggestion and as noted above, unfortunately our sample sizes are so small and the ages/sizes too similar that we do not feel our data would provide a robust analysis at this time. Those turtles that did leave the currents ranged from the smallest to some of the largest turtles we tracked (11.9 cm SCL to 18.1 cm SCL). The small number of turtles that remained in the current for the duration of their tag transmissions ranged from 13.9 cm SCL to 18.6 cm SCL. Interestingly, *Sargassum* habitat/biomass travels a very similar route as the turtles we tracked. It is also possible that some turtles were lucky to encounter *Sargassum* mats and remained within them as the mats settled into the Sargasso Sea. This is a question we are tackling as part of another study in both the Gulf of Mexico and western North Atlantic. Based on this reviewer comment, we now make note of this and their age/size suggestion in the Discussion as items to explore in the future (lines 419-424).

Line 2020: replace “codes” by ‘classes’

Done.

Line 203: Did the tracked turtles remain at the sea surface all day long? Is there actually a way to know? If not, maybe add “mostly”

We added the word ‘mostly’ here per the referee’s suggestion. They are correct in that the resolution of our data is not detailed enough to know that they spent 100% of their time at the sea surface.

Line 204: please add “and the rest were locations classes A and B”

Edit added and for consistency, also changed another reference to location code to location classes in the line above.

Line 213-217: Is it possible to show the Sargasso Sea and maybe the Gulf Stream on Figure 3?

Yes, we have updated Figure 2 to show an approximation/composite vector for the Gulf Stream and the boundary for the Sargasso Sea. We opted to include this in Fig 2 since this will introduce the region sooner in the text (and with an associated figure to illustrate the region).

Line 215 and 217: “to the east of the Gulf Stream boundary”. Do you mean to the west? For me, the east of the gulf stream is towards Europe, or do you mean the eastern boundary of the southwest part of the Gulf Stream?

Our text is correct—we are referencing the right side of the Gulf Stream or the area of

the Sargasso Sea (western S. Sea) that is just to the east of the Gulf Stream. The Gulf Stream is one of four currents that make up the NASG (we add a new figure 2a that now defines this region). It runs up along the Eastern US seaboard where it then turns to the east and becomes the North Atlantic Current (and the North Atlantic Drift) where it heads towards Europe. But where it runs up along the US coast, the eastern “bank” of the Gulf Stream serves as one of the borders to the Sargasso Sea and the western portion of the Sargasso Sea and that is what we are referencing here. We added a couple edits to this section to hopefully clarify this a little with added consistency with language used earlier (and in reference to the referee’s first comment regarding this; lines ~301-307). We are happy to add additional clarification if needed.

Line 220: please remove “were”

We kept the ‘were’ but realized that we needed to add ‘that’ before the ‘were’. This should correct the problem identified by the referee.

Line 222: Please explain how and why these 17 individuals were selected for this analysis? Why were the 10 individuals with statistically significant orientation not selected instead? I am not entirely comfortable with pooling individuals together while some of them had significant orientation and other didn’t. Please justify.

We feel it is important to have all individuals pooled for the comparison to avoid any potential concerns/perceptions about cherry-picking data to fit our objectives. However, we have added analysis for only individuals with significant heading in the revision. It made no difference in terms of the results or our conclusions (lines 322-325).

As noted above, we pooled all track data (loggerhead and green) into tracks that were south of 37° N and west of 72° W which was 17 tracks for the green turtles and all tracks for the loggerheads, also n=17. As noted above, this subset represents the spatial range within which both green and loggerhead tracks occurred, and only a few individual tracks extending beyond this range (the northern limit of the NASG). This region includes the region tested in loggerheads from Mansfield et al. [2014] encompassing the artificial magnetic field locations used by Lohmann & Lohmann [2006]—a region where laboratory-reared naive loggerheads oriented to the ENE to theoretically remain within the NASG.

Line 227-228: please replace ‘is’ by ‘was’

Correction made.

Discussion

Line 239: remove “of”

Removed.

Line 286-287: Were the turtles release at the same time of the year? If not, could that have had implications for the differences between the two species?

In both studies, turtles were lab-reared to a size that allowed us to safely put 9.5g tags on the turtles. All turtles were collected from their nests during similar hatching season

periods and held for a similar amount of time with releases staggered between December and May/June for both species. It is possible that this may tie more with annual *Sargassum* biomass availability for both species vs. within species differences...something we do not have data to examine in this paper; however, this is an excellent point that we will try to address in another (current) study...(thank you). We've added a note about this in the text (166-171, 329-225).

Line 296-299: maybe modify as : “This suggests that either (a) both species undergo an ontogenetic shift at the same age and therefore loggerheads grow faster than green turtles or they don’t and (b) green turtles spend less time in the oceanic phase of their development”

We did keep the reference to the conditions (temperature and forage availability) but added this as a third hypothesis where loggerheads and greens encounter different thermal regimes/*Sargassum*/forage availability since this ties in with some of the other edits we've included in this revision (based on this referee's review). See revised paragraph starting at line 441.

Line 299-302: this was not explicitly tested in this study. Please either remove from discussion or add corresponding results.

We edited this section to remove language suggesting we explicitly tested this and phrased things in terms of setting up new hypotheses to test with future research. Our purpose in including this is to spark new work to test for additional differences or factors that may better explain what these turtles are doing. ~Lines 364-377 (paragraph moved earlier in Discussion).

Line 304-311: This is an interesting finding. Any hypothesis as to why there is such a difference between these two species? This paragraph seems to indicate that the young loggerheads use a different part of the Sargasso Sea, is that correct? Or would you still say that they share the same nursery (as suggested in the conclusion). Maybe be more specific in what is similar between the two species and what is different and why? And if these differences are meaningful in terms of conservation or if we simply need to protect a larger area to encompass both nursery habitats?

Agreed! This is the million dollar question...It could be due to annual variation *Sargassum* biomass (a question my lab and others are trying to tackle)—with relatively small sample sizes across a limited number of years, it's difficult to say with the data used in this study. Both species use the Sargasso Sea and what we observed was that the green turtles appeared to leave the Gulf Stream sooner than loggerheads. This would imply that they may use a different part of the Sargasso Sea; however, I am hesitant to make this claim given that the relatively short transmissions of the tags in both studies—it's possible that the turtles all inhabit a similar region but the tags stopped transmitting—likely because the turtles shed the tags—before they reached any destination. So I would say in a broad regional sense, they share the same nursery region. We know that the Sargasso Sea is important for both species. Our data may be somewhat spatially biased because tags ceased transmitting and turtles might remain in that region for at least a couple of years—for which we don't (yet...) have track data. I'd

say that right now, we don't have data to really tease out any species-specific conservation measures other than to recommend that the Sargasso Sea (in particular the western portion of the Sea at a minimum) be recognized as an important sea turtle nursery for at least two species of turtles in the North Atlantic.

We edited the text in this section (Lines 409-439) to clarify and expand things based on the referee's suggestions.

Table 1: please define "body depth".

Definition and citation added to the Table 1 legend.

The term "static water" is used in the legend but not in the main results. Please make sure oceanographic descriptions are consistent throughout the manuscript.

Good catch, this was an artifact of an earlier iteration of this manuscript where we tried to quantify days in association with currents or eddies. We removed these data from the manuscript in an interest of streamlining this paper. The reference to static water and these data are now removed (lines 841-843).

Line 403: please change "to enter" by "entering"

We assume the reviewer meant to refer to line 503 and not 403? On line 503, we changed "to enter" to "entering".

Figure 4 , Line 517: I thought the orientation headings were calculated every 6 hours and not daily. Please check with methods and modify if necessary.

The headings were calculated every 6 hours first. Then, we calculated the average daily heading, which is stated in the method section. We further clarified these methods in the revised version of the manuscript. Line 236.

Table 2.a.: the U2 score for the first turtle is missing.

Thank you for catching this. We've corrected this typo and updated Table 2 to reflect additional analyses per Reviewer #2's suggestions below.

Referee: 2

Comments to the Author(s)

This study focuses on satellite tracking of young green turtles tagged off the eastern coast of the USA. It is generally well written and presents novel movement information for a life-stage that remains largely unknown for most sea turtle species and populations, thus represents a wonderful contribution that I will be excited to see in print.

Thank you.

Notwithstanding this perspective, I have several suggestions that I think would improve the manuscript, the most important of which relate to the figures:

- **Include outlines or some type of shading to show the general areas of the 1) Gulf Stream and 2) Sargasso Sea. These areas are discussed throughout the manuscript, are described as discrete areas, and analyses are conducted on locations within them, but they are not depicted anywhere. Would be a huge improvement to provide a visual of their general location/boundaries.**

Thank you for this suggestion. We've included an additional sub-figure in figure 2 (Fig. 2a)

- **Zoom in on the tracks presented in Figure 2 and Figure 3 so more detail can be seen on individual tracks. Particularly relevant when citing particular tracks (e.g., lines 190-193) If authors would like to show range of tracks within Atlantic, I suggest using an inset to do so (i.e., zoom main image in on tracks, then place a small inset in the one of the corners depicting tracks within entire Atlantic).**

We have updated these figures to be more “zoomed in” to hopefully help the reader see the details of the tracks. We are also open to providing a closer look at individual tracks in supplemental material (as examples of individual tracks) as needed/per the Editor's request.

- **Lines 213-215 – I agree that the kernel analyses are greatly biased by their initial release location and the Gulf Stream. I think doing an analysis for only the locations outside the Gulf Stream would provide a better visual for the importance of the Sargasso Sea. Suggest authors try this.**

We included an additional analysis with just those tracks occurring within the Sargasso Sea as suggested by Reviewer #2 (updated Methods lines 224-232, Results lines 301-307). We modified Figure 3 to include an analysis of only those tracks in the Sargasso Sea in addition to the original tracks (Fig. 3a-f). We also broke out the data into green, loggerhead, and green & loggerhead combined in each data grouping (all track data, Sargasso Sea track data only). We opted to keep our original analysis with all track data, including the initial release and dispersal in the Gulf Stream because we did not want to discount and remove those turtles that did not leave the currents from the analysis and did not want to give the impression that our data only represent turtle tracks that entered the Sargasso Sea (or that that is the only place these turtles occur). We think that the new analyses and break-down of the data will provide US and international species managers different and better options to work with now. Our companion paper for loggerheads – Mansfield et al. 2014—was used to establish critical *Sargassum* habitat for turtles in the US Exclusive Economic Zone which includes Gulf Stream waters. We anticipate the data in this manuscript to similarly inform oceanic critical habitat for green turtles, thus we opt to include all data in addition to the new analyses suggested by Reviewer 2.

- **Adding labels to the actual maps in Figure 3 that indicate the species included in each pane would facilitate interpretation.**

These labels are added to the new figures (all).

- **Adding labels to the rose diagrams in figure 4 would also facilitate interpretation.**

We added labels to the (new) rose diagrams (all).

However, this figure is also confusing. Line 219-221 discussed data from 10 and 11 turtles for which orientation analysis was significant and non-significant, respectively, then references Fig. 4a and b. However, Fig. 4a and b are for individual “example” turtles (at least that’s my understanding from reading the figure description), so this doesn’t seem appropriate. Why showing “example” turtles for Fig. 4a, b, d, and e? Why not showing combined results for the groups of green and loggerhead turtles? Could be misinterpreted as cherry picking the results. Suggest clarifying this figure and how it corresponds to the text.

We removed the individual rose-diagrams in this figure (4), and added the pooled data comparison for 10 green turtles and 12 loggerheads with significant heading (new Fig 4c-d). Each panel is now labeled to clarify species and all or subsampled datasets). Methods lines 242-254, Results lines 309-325.

- **Because this study really aims to compare and contrast green and loggerhead tracks, would recommend a map that shows the tracks for both species, I would also consider including a map that contains tracks from both species/studies.**

To Figure 2, we added an adaptation of our map we published in the Mansfield et al. 2014 paper per this reviewer’s request (see new Fig. 2c). We, of course, give ourselves permission to use our data to recreate this map (to the Editor: please let us know if we need to get journal consent for this as well? It is the same journal and the map is different in scale).

The manuscript states that all sea turtle species (except flatbacks) migrate from natal neritic waters to oceanic nursery areas, then recruit to neritic area at a certain size (Line 50-52 & 324-326). Although I would agree that some (most?) populations of all species may undergo this movement pattern, I’d argue that the jury is still out on whether all populations of all species do. For instance, tracking research of young-of-year hawksbill turtles that I am involved with (currently in the process of being written up by Dr. Mike Liles) has found that many of the turtles migrate directly into the estuarine waters adjacent to the beaches where they were born. By stating that offshore migrations is a given, I think the authors actually undercut the novelty of their study (and the previous Mansfield manuscript), which are really the first to demonstrate this actually occurs (rather than simply being a hypothesis) and provides justification for why much more of this type of research is needed. I recommend leaving this as behavior that has been confirmed for the two species/populations in question, but remains an open question for many other populations.

Fair point and we strongly agree that as we learn more, these old hypotheses cannot be upheld for all turtles in all areas. The section in question is more of a review of the available literature; however, we agree that our previous and pending new work (and

the exciting work that reviewer notes is in preparation) all point to the fact that the old assumptions and hypothesis can't be applied to all turtles in all places. We've edited the text in the Background to better reflect this (noting that this is an assumption) and to set up our argument for this in the Discussion (and deleted the text referencing flatbacks in the discussion).

I recommend the authors consider rewriting much of the first paragraph in the discussion (specifically lines 237-248). The authors start by contrasting the initial beaches/habitats of green and loggerhead turtles in the study areas. I was expecting this to transition into how they tend to use different oceanic areas for development, i.e., lines 308-310 seem to be the more relevant take-home message and I think it would be more effective to make this point (condensed version) at the start of the introduction (i.e., could still leave lines 308-310 as they are). However, rather than doing that, the first paragraph goes into the morphology of the two species and discusses the different tag adhesion methods, which would seem more appropriate for a "methods" manuscript. I recommend restructuring the opening discussion paragraph. Consider a separate section somewhere else discussing tag adherence, longevity, etc., or omit that part altogether.

Our intent with the order of the Discussion is to follow with the order in which we present our Methods and Results. Given that this is a new technique for tagging small green turtles, we think that it is important to include this in the Discussion as it will hopefully inform future researchers' techniques. However, we reworked most of the Discussion to include a broader introductory paragraph, then a break-down following the flow of the other sections in the manuscript (tag-related details, then movements, and final conclusions related to species conservation). We also shifted the order of some paragraphs to allow for better flow. Hopefully, these edits will address Reviewer #2's concerns and/or clarify our intent.

Additional comments:

Line 29 – this assumes sea turtles undertake movements over 1,000's of kilometers across open ocean. This isn't necessarily always the case and certainly hasn't been shown for many species or populations (hence the novelty of this study).

We see the referee's point and are not stating that *all* turtles travel 1000s of km, but we argue that some young turtles do indeed travel 1000s of km in the North Atlantic and South Atlantic (Mansfield et al. 2014; 2017; Putman and Mansfield 2015). Also early work examining the mtDNA of turtles captured off the Azores, Cape Verde, and Canary Islands link those turtles to US east coast rookeries. And aside from direct evidence, many publications suggest/hypothesize/assume that North Atlantic oceanic stage turtles migrated around the Atlantic during their first years at sea. See Carr 1986, 1987; Bjorndal et al. 2000; Bolten et al. 1993, 1998. Unfortunately, the abstract word count does not allow for a full justification for our original statement to be added. As a compromise we edited the sentence to remove the specific 1000 km note and to better reflect the need for and challenge of tracking young turtles. Lines 29-30.

Line 30 – "Long-term" and "very young", the former used in several parts of the

manuscript, are quite ambiguous terms. I would try to be more specific (or define what you consider “long-term” and why).

The abstract word count limits our ability to address this request in full here. We therefore clarify the use of ‘long-term’ and ‘young’ in the introduction and elsewhere. We use the term ‘young’ interchangeably with oceanic stage and add additional clarification in the introduction. We edited out the term ‘young’ from the abstract since the word count does not allow additional clarification. We also use the term ‘young’ more sparingly throughout the manuscript and provide reference for size or age of turtle when appropriate.

Line 36 – Remove “of”

Removed.

Line 48 – There are multiple definitions for the term “oceanic” so it might be worth defining it here as relating to the open-ocean just to be clear. Line 81 says, “(remaining off of the Continental Shelf or in waters >200 m depth)” ...is that what the authors consider oceanic?

Yes, we use the standard oceanographic definition of ‘oceanic’ as waters >200m in depth. Additional reference to the >200m definition are added to the background section. Line 54.

Line 53 – what is the “L.” after *Chelonia mydas*?

The L is the taxonomic authority that named the species. In this case it refers to Linnaeus who first described the species. This is the proper way in which scientific names are written. We realized that we are not consistent with this when other species are mentioned and deleted it (some journals require this added initial). Line 59.

Line 68 – Lines 253-254 in the discussion section define the Sargasso Sea. I suggest moving that description here.

Suggested edit was made. Lines 75-78.

Line 68 – define ENE.

ENE is now spelled out in the text. Line 96.

Line 72-74 – Suggest providing an example(s), or at least a citation(s).

Citations added. Line 63.

Line 78 – “Turtles in this study” is a bit of an unclear reference. I initially thought the authors were referring to turtles in the current study and it threw me off. Suggest tweaking to make clear.

Tweaked to specify loggerhead turtles and added an additional citation of the work. Line 111.

Line 80 and 251 – “long standing hypotheses” by who? Provide a reference(s).

Please see edits in response to Reviewer 1 (and the editor) where we spell out these hypotheses earlier in the Background and provide citations. We edited the text

referenced here to include direct reference to the specific hypotheses and have also added the relevant citations. See lines 71-82, as well as sections throughout the rest of the Background.

Line 96 – Given Mansfield is the first author of this manuscript, I would suggest rewording “Building on the work of Mansfield et al. [19, 20, 22]...” to “Building on previous studies (Mansfield et al. [19, 20, 22])...”

We made this edit in the text. Line 147.

Line 133 – what constitutes “adapted well”?

We edited this statement to note that smaller turtles did not meet our attachment criteria, but turtles >300g did, so we used green turtles >300g in this study. Lines 188-195.

Line 161 – Missing closing parenthesis after citations.

Good catch, thanks. Edit made in the text.

Line 170-171 – Describe how/why was this geographic subset chosen. I know it was done to test differences in mean orientation, but explain why tracks within this geographic area.

Reviewer 1 (above) asked a similar question and here is our response: this subset represents the spatial range within which both green and loggerhead tracks occurred, and only a few individual tracks extend beyond this spatial range (the northern limit of the NASG). The subset of locations includes the region tested in loggerheads from Mansfield et al. [2014] encompassing the experimentally-generated magnetic field locations used by Lohmann & Lohmann [2006]—a region where laboratory-reared naive loggerheads oriented to the ENE to theoretically remain within the NASG. We made associated in-text edits to clarify this subset (see response to Reviewer 1 for specific lines in text).

Line 190-193 – I cannot distinguish the referenced tracks in Figure 2 (see my previous comment regarding zooming in for this figure).

We have “zoomed in” on the smaller geographic range, the tracks should now be clearly visible. Please see revised Figures 2b and c.

Line 194-199 – These numbers don’t seem to coincide/add up. First says 15 of the 21 turtles departed the Gulf Stream, then says eight departed south of Cape Hatteras and eight departed north/east. Also, $8+8+3$ (that had not entered the Sargasso Sea prior to tag cessation) = 19, so what about the other two turtles. Please make this clearer.

Yes, thank you. Reviewer 1 also caught this typo and we’ve corrected the error in the text. See response to Reviewer 1 above for the lines associated with this edit.

Line 202 – It would seem prudent to mention something in the methods about how tag charging data is collected.

We are unsure if the reviewer wants a technical explanation of the tag function or simply clarification that charge at time of transmission is one of the data points collected by the tags and transmitted as a sensor output? We opted for the simple explanation since the data are part of the tag output and not something we can change or modify within the tag function for experimental purposes. An additional mention was also made in the Methods section. Line 214.

Line 203 – reads as if tracked turtles ALWAYS remained at the surface, but cannot be sure.

Agreed – in response to Reviewer 1, we added the word ‘mostly’ here. Line 290.

Line 205 – suggest moving to methods

We opted to keep these data reported in the Results section because we don’t report the track data until the Results. The LCs are important results that also help our argument that turtles are likely remaining at the sea surface most of the time (as evidenced by the higher accuracy LCs).

Line 206 – Add “V” after 4.04.

Added, thanks for catching that omission. Line 295.

Line 285 – provide the proportion (%) of loggerheads that departed the Gulf Stream in the previous manuscript.

The number (7 of 17) was added to the text here as well as a reminder of the number of green turtles leaving the Gulf Stream. Line 284-286.

Line 304-305 – seems that this sentence should proceed the previous paragraph.

In response to Reviewer 1, we’ve substantially edited this part of the discussion (including the paragraphs above). The point of this paragraph is related to the new hypotheses posed in the paragraph above it (within the text). We added edits to this section to improve the flow (and relevance) and hopefully address Reviewer #2’s concerns.

Lines 323-324 – the purpose of including the portion of the sentence, “no sea turtle species remains exclusively within US waters throughout their long lives” is unclear to me. Elaborate.

The point of this was to note that turtles originating from US beaches are not exclusively “US turtles” remaining in US waters throughout their lives. We edited the text to better reflect our intent. Line 481-483.

Congratulations on a great manuscript and I hope my comments are useful.

Thank you and yes, very helpful. We think that the paper is improved and we are very excited to see your hawksbill paper in press – please send it along when it is available (feel free to list me as a reviewer). Cheers, Kate

**Kind regards,
Alexander Gaos**